# Developmental progression of DNA double-strand break repair deciphered by a single-allele resolution mutation classifier

Zhiqian Li[1,2], Lang You[1,2], Anita Hermann[1,2] & Ethan Bier ®[1,2] ✉

DNA double-strand breaks (DSBs) are repaired by a hierarchically regulated network of pathways. Factors influencing the choice of particular repair pathways, however remain poorly characterized. Here we develop an Integrated Classification Pipeline (ICP) to decompose and categorize CRISPR/Cas9 generated mutations on genomic target sites in complex multicellular insects. The ICP outputs graphic rank ordered classifications of mutant alleles to visualize discriminating DSB repair fingerprints generated from different target sites and alternative inheritance patterns of CRISPR components. We uncover highly reproducible lineage-specific mutation fingerprints in individual organisms and a developmental progression wherein Microhomology-Mediated End-Joining (MMEJ) or Insertion events predominate during early rapid mitotic cell cycles, switching to distinct subsets of Non-Homologous End-Joining (NHEJ) alleles, and then to Homology-Directed Repair (HDR)-based gene conversion. These repair signatures enable marker-free tracking of specific mutations in dynamic populations, including NHEJ and HDR events within the same samples, for in-depth analysis of diverse gene editing events.

DNA double-strand breaks (DSBs) can be generated by intrinsic cellular processes such as transcription, replication, or by external DNA damaging agents, including chemicals and irradiation. Such DNA lesions pose immediate threats to genomic integrity, and failure of DSB repair underlies many human diseases such as tumorigenesis, cancer, and cell death due to accumulation of deleterious lesions and the generation of genomic instability[1–4]. Unicellular and multicellular eukaryotic organisms have evolved sophisticated hierarchical networks of DNA repair systems to resolve DSB lesions, mediated by two broad primary categories of corrective pathways often referred to as Nonhomologous End-Joining (NHEJ) and Homology-Directed Repair (HDR)[3,5,6]. The former, NHEJ, which acts throughout the entire cell cycle, directly reconnects loose ends with no involvement of DNA repair template (canonical NHEJ or c-NHEJ). If the DNA target is subject to recurring cleavage, however, as can result from persistent exposure to sequence-specific nucleases, errors may eventually arise leading to production of cleavage resistant mutations. By contrast, HDR is

predominantly active during late S and G2 phases of the cell cycle and resolves DSBs by gene conversion using exogenously provided homologous DNA, a sister-chromatid, or the homologous chromosome as the repair template[7,8]. A DSB repair decision tree determines the selection of NHEJ versus HDR pathways for resolving a given DSB lesion[3,9]. This binary DSB repair choice is oversimplified, however. Identification of new repair pathways and overlapping mutational signatures generated by distinct repair processes such as Microhomology-Mediated End-Joining (MMEJ), which is highly active during mitosis, and Single-Strand Annealing (SSA) underscore how additional repair outcomes need to be considered[6,10–13].

Mechanistic models of DSB repair have been informed by foundational studies performed in diverse species of metazoans by treating simple model systems including budding yeast and mammalian cells with physical or chemical DNA damaging agents as well as through genetic analysis in yeast and *Drosophila* in response to radiation induced mutagenesis or site-specific DNA breaks induced by

[1]Department of Cell and Developmental Biology, University of California, San Diego, La Jolla, CA 92093, USA. [2]Tata Institute for Genetics and Society, University of California, San Diego, La Jolla, CA 92093, USA. ✉e-mail: ebier@ucsd.edu

endonucleases including *I-SceI*, zinc finger nuclease, TALENs, and CRISPR[6,14–20]. In the case of site-specific DNA damage induced by CRISPR/Cas9 in mammalian cell lines, much of this analysis has been conducted with exogenously provided DNA repair templates or, in a few instances, using the sister chromatid or homologous chromosome as repair templates[21–27]. These studies often employ quantifiable fluorescence reporters to track and quantify different repair outcomes including: HDR[28], NHEJ[29], MMEJ[30], and SSA[31,32]. A limitation of many such studies, however, is the infeasibility of testing multiple loci in different cell types and distinguishing how alternate repair pathways contribute to diverse repair outcomes of intact complex developing organisms[33]. High-throughput next generation sequencing (NGS) combined with custom developed bioinformatic pipelines have overcome some of these limitations opening new avenues for characterizing factors that influence DSB repair pathway choices[34–37]. Recently, a sophisticated DSB repair classifier system was developed to map the genetic landscape of DSBs at high resolution, enabling a detailed analysis of the usage of particular pathways in stereotyped repair outcomes[9]. Nonetheless, these and other analytic tools amenable for tracking simple editing outcomes are not typically designed for comprehensive characterization of both gene conversion mediated HDR events nor for classifying diverse mutations such as those generated by the NHEJ or MMEJ pathways within the same sample[38,39]. Nearly all current DSB classifier systems assess DNA outcomes in homogeneous cell types such as cultured cell lines, leaving open what role final diverse cell fates or those arising during development may play in determining editing outcomes. Therefore, analyzing and classifying DNA repair outcomes at diverse native genomic DNA sites at fine scale with single-allele resolution within complex tissues composed of different cell types remains a challenging objective. Similarly, powerful single-cell DNA sequencing methods, which have been translated in analyzing and categorizing cell-type specific programs, are technically limited in scope when applied to analysis of DSB repair of a specifically targeted genomic DNA locus[40].

Here we apply a newly developed highly discriminating mutation classifier system, the Integrated Classifier Pipeline (ICP), to decompose and categorize Cas9 induced DSB repair outcomes in complex multicellular organisms with single allelic resolution. This ICP pipeline is particularly revealing in that it outputs intuitively displayed rank-ordered and sub-categorized mutational allele fingerprints, rather than specific primary DNA sequences. This higher-order classification of mutations distinguishes remarkably reproducible and defining alternative categories of DNA-repair outcomes in somatic cells of individual flies and mosquitoes that depend on different target sites, alternative inheritance patterns of CRISPR components, and alternative repair pathway usage based on developmental stage. The discriminating nature of ICP outputs also enables marker-free tracking of specific mutations in dynamic freely mating populations and permits simultaneous quantification of both NHEJ and HDR events within the same sample. The ICP platform offers particular future advantages to surveillance of gene-drive performance in insects and potentially to more discriminating assessments of off-target effects in diagnostic gene therapy and other broad gene-editing contexts.

## Results

### ICP: an integrated pipeline for classifying CRISPR/Cas9 induced mutant alleles

We developed an integrated bioinformatic tool ICP (Integrated Classifier Pipeline), to parse complex DSB repair outcomes induced by CRISPR/Cas9 and automatically call for experimental errors generated during NGS library preparation and sequencing: **1)** a Nucleotide Position Classifier (NPClassifier), and **2)** a Single Allele-resolution Classifier (SAClassifier). We employed these two complementary sequence analysis modules in tandem to enable in-depth interpretation of deep sequencing data at single allele resolution (Fig. 1a–c, see Methods

section for detailed description of ICP tools). In line with the unique DNA signatures generated by distinct DSB repair pathways, we categorized the repair products into four major categories. Alleles with a deletion only on the PAM-distal side (PAM-proximal side was protected by Cas9 protein after cleavage), a common category, were termed as PEPPR class mutations (PAM-End Proximal Protected Repair, PEPPR)[41,42]. While single strand cleavage by the Cas9 RuvC domain can also nick the non-complementary strand at locations beyond the canonical site between the 6th and 7th nucleotide upstream of the PAM sequence, we restrict our analysis here to the majority cases wherein Cas9 cleavage generates blunt DSB ends to simplify the robust classification scheme developed in this study[43–45]. Mutant alleles judged to be generated by directly annealing ≥2 bp microhomology sequences spanning the gRNA cleavage site were assigned into MMEJ class (again acknowledging that such alleles can also be generated with 1 bp microhomology sequence, which however, are not readily amenable to the semi-automated analysis we developed)[46–48], while pure deletion alleles not belonging to either the PEPPR or MMEJ categories were classified as DELET class mutations. Remaining alleles that include insertions-only and indels (deletion plus insertion) were categorized as insertion class (INSRT) mutations (Fig. 1b).

Briefly, raw reads generated from deep sequencing were subjected to a preliminary categorization using the NPClassifier, which recognizes the relative positions of editing start- and end-points flanking Cas9 cleavage site and then generates a collection of *priori* alleles for each category. These primary outputs (MMEJ and DELET) were used for building full-length standard comprehensive dictionaries listing all observed mutations and derived 24-nt short dictionaries (with the same seed region flanking the Cas9 cleavage site) as inputs of the SAClassifier. In addition, a synthetic PEPPR dictionary was built by iteratively increasing the length of deletions by a single nucleotide distal to the PAM site, excluding alleles belonging to the MMEJ category. By fishing the raw reads with 24-nt dictionaries, we were able to automatically recognize reads that also contained experimentally generated errors (e.g., from PCR amplification), which usually are located outside of the narrow 24-nt short dictionary window, thereby assigning such composite alleles to correctly matched root alleles (Fig. 1b). These dual iteratively employed ICP classification tools provide a robust and precise classification of CRISPR/Cas9 induced DSB repair outcomes. Next, we developed an evocative user-friendly interface to visualize processed allelic category information in the form of rank ordered allelic landscape plots and repair pattern fingerprints (color-coded DSB repair categories), both of which are sorted by read frequency (Fig. 1c). These intuitively accessible data outputs are far more informative and discriminating than the unprocessed primary DNA sequence reads (e.g., compare the seemingly idiosyncratic raw lesions depicted in Fig. 2a to the obviously unique processed and concordant replicate patterns shown Fig. 2b, c). The ICP was thus employed to visualize results in all the following experiments.

### Highly reproducible and specific allelic fingerprints are generated by alternative CRISPR/Cas9 inheritance patterns

Since DSB repair outcomes have been found to vary considerably as a function of Cas9 or gRNA source and level[49,50], we employed the ICP platform to parse somatic indels generated by co-expressing Cas9 and gRNAs in somatic cells of fruit flies (*Drosophila melanogaster*) and mosquitoes (*Anopheles stephensi*) in various configurations associated with gene-drive systems. We first applied ICP analysis to a split gene-drive system inserted into the *Drosophila pale* (*ple*) gene that is designed to detect copying of a gene cassette in somatic cells. This element, referred to as a CopyCatcher (*ple*CC), carries a gRNA targeting the first intron of *Drosophila ple* locus[49]. In this current study, we make use of low-level ectopic somatic Cas9 expression (which is substantial and broad for *vasa*-Cas9) to analyze DSB repair patterns across diverse cell types in F$_1$ progeny carrying both Cas9 and gRNAs[51–53].

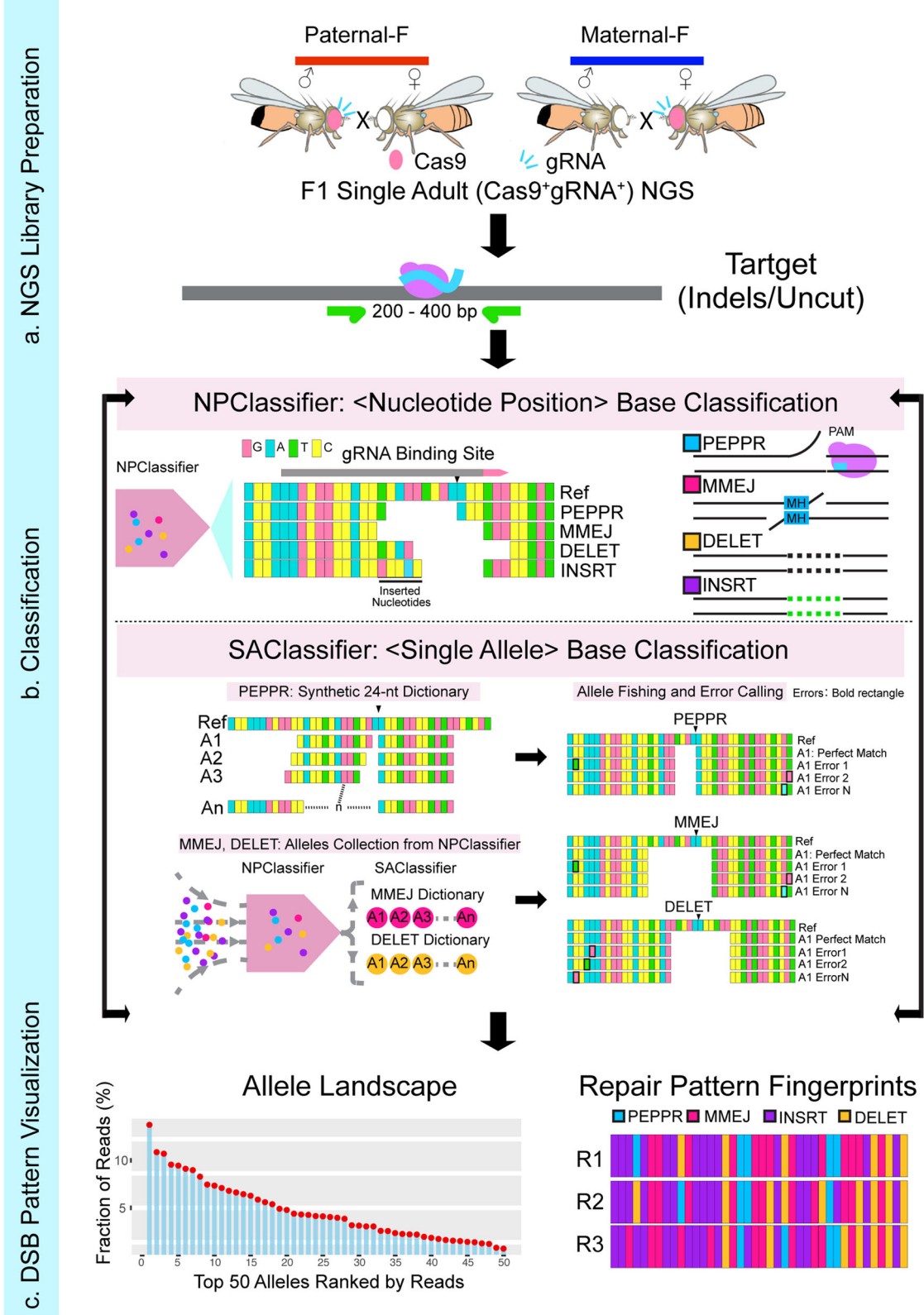

Because cells actively undergoing meiosis make up only a small fraction of dividing cells in an adult fly, the mutational effects of Cas9/gRNA cleavage in such $F_1$ individuals largely reflect the somatic action of these nuclease complexes. We thus conducted several alternative crossing schemes to assess the somatic mutagenic activity of *vasa*-Cas9 and gRNA components when transmitted to $F_1$ individuals in various configurations from their $F_0$ parents: **1)** Maternal Split (Maternal-S, females carrying *vasa*-Cas9 crossed with males carrying *ple*CC); **2)** Paternal Split (Paternal-S, males carrying *vasa*-Cas9 crossed with females carrying *ple*CC); and **3)** Maternal Full (Maternal-F, females carrying both the *ple*CC and *vasa*-Cas9 transgenes); or Paternal Full (Paternal-F, males carrying both the *ple*CC and *vasa*-Cas9 transgenes)[49]. Comparative ICP analysis revealed several striking and consistent differences between the prevalent somatic mutations

**Fig. 1 | Design and workflow scheme for using the ICP platform to parse CRISPR/Cas9 induced DSB repair outcomes.** The process of DSB repair pattern profiling consists of preparing a NGS library (**a**), classifying the resulting parsed alleles (**b**) and displaying processed alleles by rank order and class of mutations (**c**). **a** NGS library preparation: Genomic DNA from F$_1$ test flies carrying both Cas9 and gRNA expressing cassettes either maternally (dark blue bars) or paternally (red bars, or progeny from other designated crosses) are subjected for targeted PCR amplification with primers containing Illumina compatible adapters at the 5′ terminal to detect somatic indels. The gray rectangle represents a short region of genomic DNA containing a Cas9/gRNA target: purple circle depicts Cas9 protein and sky-blue line is gRNA. **b** Classification: Raw NGS data are subjected to the NPClassifier to parse alleles into specific primary categories required for building allelic dictionaries used by the SAClassifier. Four major indel groups are categorized: PEPPR (PAM-End Proximal Protected Repair, sky-blue), MMEJ (Microhomology Mediated End-Joining, dark pink), DELET (deletion, any deletions do not belong to PEPPR and MMEJ, orange) and INSRT (insertion, including the alleles only with inserted nucleotides or had deletions and insertions, purple). The 24-nt short PEPPR, MMEJ and DELET dictionaries are used for a more accurate classification and error calling by binning together all alleles with the same seed region that match primary allelic entries in the SAClassifier dictionaries. **c** DSB repair pattern visualization: intuitive rendering of the processed raw sequence data as an output of rank ordered classes of alleles. Allelic classes derived from NGS sequencing of individual flies or mosquitoes are displayed by their ranked frequency (allele landscape) and repair pattern fingerprints (color-coded by categories).

generated in individual progeny in each of these different crossing schemes. In the case of Paternal-S crosses, the resulting mutations were dominated by PEPPR alleles (4 out of top 5 alleles in Fig. 2a, Fig. S1a, and 70% of the top 50 alleles as rendered in rank ordered allelic landscapes and color coded DSB repair fingerprints in Fig. 2c). In contrast, Maternal-S crosses primarily generated MMEJ and INSRT indels (4 out of top 5 alleles were MMEJ, and at least 50% of the top 50 alleles were INSRT mutations, Fig. 2a, c, Supplementary Fig. S1a). These differences were also evident in the steeper allelic landscape curves that were generated from the Maternal-S versus Paternal-S crosses (Fig. 2b) as characterized by the initial portion of the curve depicting the 5 most frequent alleles (i.e., the dark blue lines in Fig. 2b are all above the red lines for the 5 most frequent alleles). We further quantified differences in allelic profiles between crosses by bar plots displaying the summed proportions of the different allelic classes (summing the percentages of all alleles from each category) which we termed as Class Fraction (Fig. 2d). This analysis revealed that INSRT alleles were generated at a significantly higher frequency in Maternal-S crosses, while the PEPPR class dominated among the top 50 alleles in the reciprocal Paternal-S crosses (Fig. 2d).

A striking feature of the highly divergent DSB repair signatures generated from maternally versus paternally inherited Cas9 sources was the remarkable reproducibility of their DSB repair fingerprints observed across three individual replicates from each cross (Fig. 2e, f). We performed a correlation analysis within replicates by extracting 23 common alleles across all six sequenced flies and plotted the resulting allelic profiles together relative to an arbitrarily chosen Paternal-S replicate as reference (bold red line, Supplementary Fig. S1b). We observed that the frequency distributions of these 23 common alleles were much more similar to each other within intra-cross comparisons than between inter-crosses (Supplementary Fig. S1b). This trend was also revealed by higher correlation coefficients for intra-cross comparisons than for inter-cross comparisons based on allelic read ratios (Supplementary Fig. S1c–g). Conspicuous defining differences between the Maternal-S and Paternal-S fingerprints were also evident based on the Class Fraction index (Fig. 2d). In summary, a variety of differing statistical measurements all underscore the robust consistent similarities shared among allele profiles generated from individual replicates of same cross and clearly distinctive DSB repair pattern fingerprints generated by maternal versus paternal Cas9 inheritance.

We extended our ICP analysis of mutant allele profiles generated in the *ple* locus to the more extreme Maternal-F (dark blue lines) and Paternal-F (red lines) cross schemes to assess the role of inheritance patterns when both the source of *vasa*-Cas9 and gRNA originated from a single parent[49]. Again, we observed highly dominant alleles in the Maternal-F crosses, clearly evident in allelic landscapes, that deviated markedly from those produced by the Paternal-F crosses, which produced more evenly distributed spectra of alleles spread across a broad range of allelic frequencies (Fig. 3a, b). As expected based on these large differences, the repair pattern fingerprints generated from different crosses produced clearly distinguishable patterns of mutation classes, which was particularly evident when considering the Class

Fraction (Fig. 3e). Cumulatively, these data suggest that the developmental timing and/or levels of Cas9 expression (maternal, early zygotic, or late zygotic) are likely to play a key role in determining which particular DSB repair pathway or sub-pathway is engaged in resolving DSBs.

## Highly reproducible distinct DSB fingerprints are associated with different Cas9 sources

Previous studies have shown that the relative frequencies of NHEJ versus HDR events depend on the source of Cas9 both in terms of timing and level of expression[49,50,54]. We thus wondered whether ICP analysis would similarly reveal distinct DSB repair outcomes for two additional Cas9 sources (*actin*-Cas9 and *nanos*-Cas9, expressing level of Cas9: *actin*-Cas9 > *vasa*-Cas9 > *nanos*-Cas9) inserted at the same locus with *vasa*-Cas9 (Fig. 3c, d)[49].

As was observed for the *vasa*-Cas9 source, the *actin*-Cas9 and *nanos*-Cas9 sources both generated differing allelic landscapes and repair pattern fingerprints when transmitted maternally versus paternally, which also were readily distinguishable from each other (Fig. 3b–d). Mirroring results with the *vasa*-Cas9 source, significant differences between the proportions of PEPPR versus MMEJ class among the top 20 alleles were observed in Maternal-S versus Paternal-S crosses for *actin*-Cas9. For the *nanos*-Cas9 source, both the MMEJ and INSRT categories were particularly reduced in Paternal-S crosses, although this latter sex-based difference was not as dramatic as for the other Cas9 sources (presumably due to its more germline restricted expression, Fig. 3d)[55,56]. Overall, the general trend once again indicated that maternally inherited Cas9 sources biased somatic DSB repair outcomes in favor of MMEJ and INSRT classes over PEPPR alleles, while paternal transmission of Cas9 generated mutant alleles dominated by PEPPR class alleles (Fig. 3e).

Based on the overall similarities of the DSB repair outcomes observed for *actin*-Cas9 and *vasa*-Cas9 crosses, we extracted a set of 59 shared alleles that appeared in all sequenced samples and performed UMAP (Uniform Manifold Approximation and Projection) analysis to cluster these common alleles, condensing them into 5 distinct clouds (Fig. 3f). Clouds 1, 2, 3, and 4 were dominated by alternative subsets of PEPPR alleles distinguished primarily by the length of deletion (the average deletion sizes were 24 bp, 40 bp, 31 bp for PEPPR Mini, Midi-I and Midi-II cluster, and it was longer than 55 bp for PEPPR Maxi cluster), while cloud 5 was predominantly comprised of MMEJ alleles. We reviewed raw sequences for the few trans-cloud assigned alleles and discovered that some of these alleles could be interpreted as having been generated from a second round of repair using one of the core alleles from the same cloud as a repair template. For example, we inferred that allele 58 was actually a PEPPR deletion with several nucleotides potentially having been back-filled. This result is consistent with the previous report that alleles with insertions or complex repair outcomes would be generated from several rounds of synthesis following the generation of a primary deletion event[57,58]. Assessing the impact of such potential complexities,

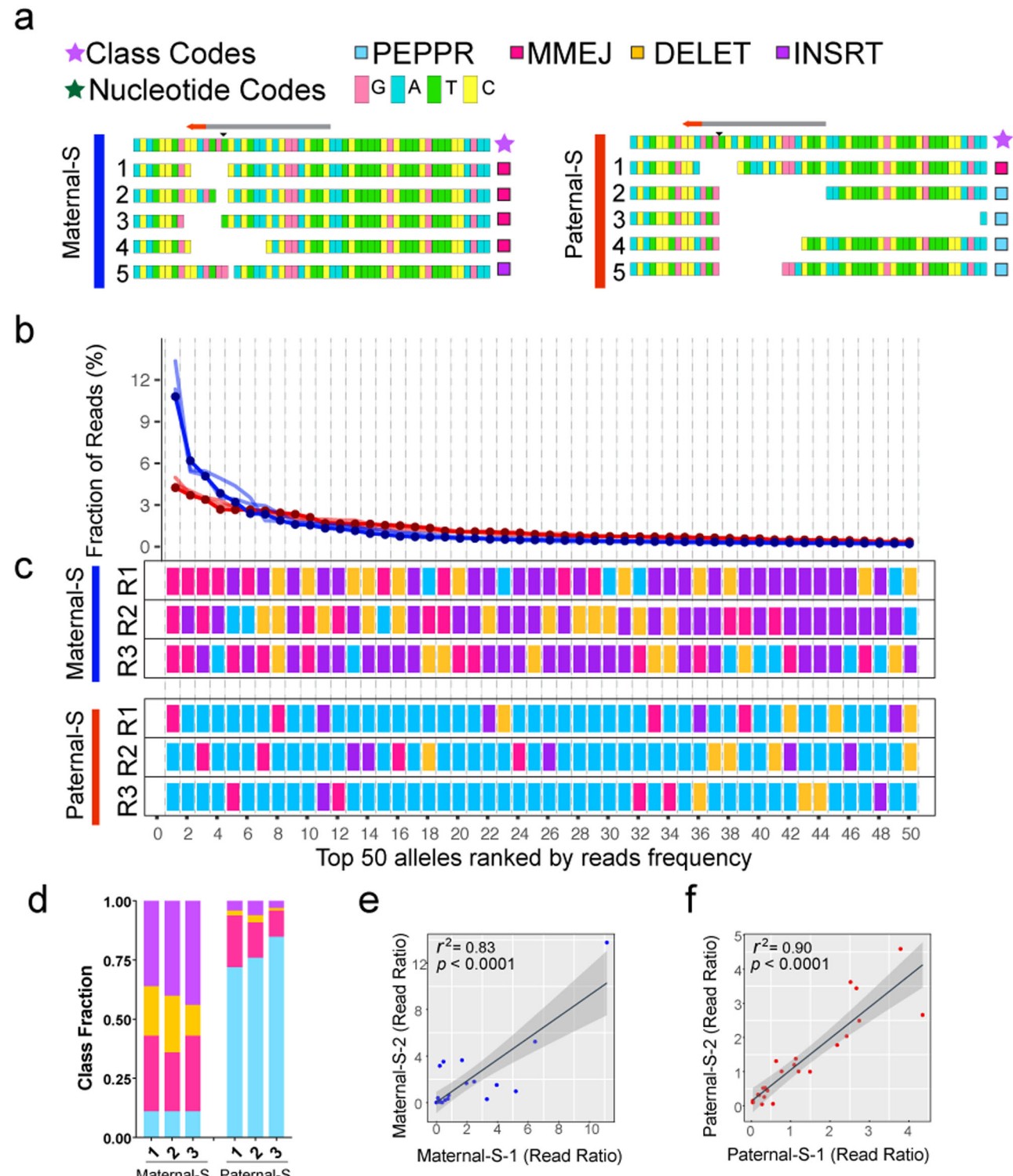

**Fig. 2 | The ICP resolves distinctive DSB repair fingerprints at the *Drosophila pale* locus. a** Examples of the top five somatic indels from individual flies derived from split-drive crosses in which the Cas9 transgene is inherited either maternally (Maternal-S, left) or paternally (Paternal-S, right), but separately from a cassette carrying the gRNA transmitted by the other parent. Purple stars indicate the color codes for mutation categories (dark pink: MMEJ, sky-blue: PEPPR, orange: DELET, purple: INSRT) and dark green star indicates the separate raw sequence color coded for the four nucleotides A, T, G, and C. The red bar indicates Paternal-S crosses while dark blue bar represents Maternal-S crosses. **b** Landscapes of top 50 alleles ranked by reads ratio. All six sequenced individual flies are plotted together, with dark blue lines plotting the data from Maternal-S crosses and the red lines from Paternal-S crosses. The y-axis presents the fraction of reads for a given allele and the x-axis depicts the top 50 alleles according to rank order by read frequency. **c** DSB repair fingerprints for three representative sequenced individual flies from each cross. The x-axis is the same as depicted in panel **b**. Both panels show the top 50 ranked alleles. **d.** Bar plots of Class Fraction for top 50 alleles. Color codes for classes are as in panels **a** and **c**. Correlation analysis of two out of three replicates from Maternal-S cross (**e**) or Paternal-S (**f**) cross. $r^2$ values and *p*-values are indicated. Source data for panels **b**, **d**, **e** and **f** are provided as a Source Data file.

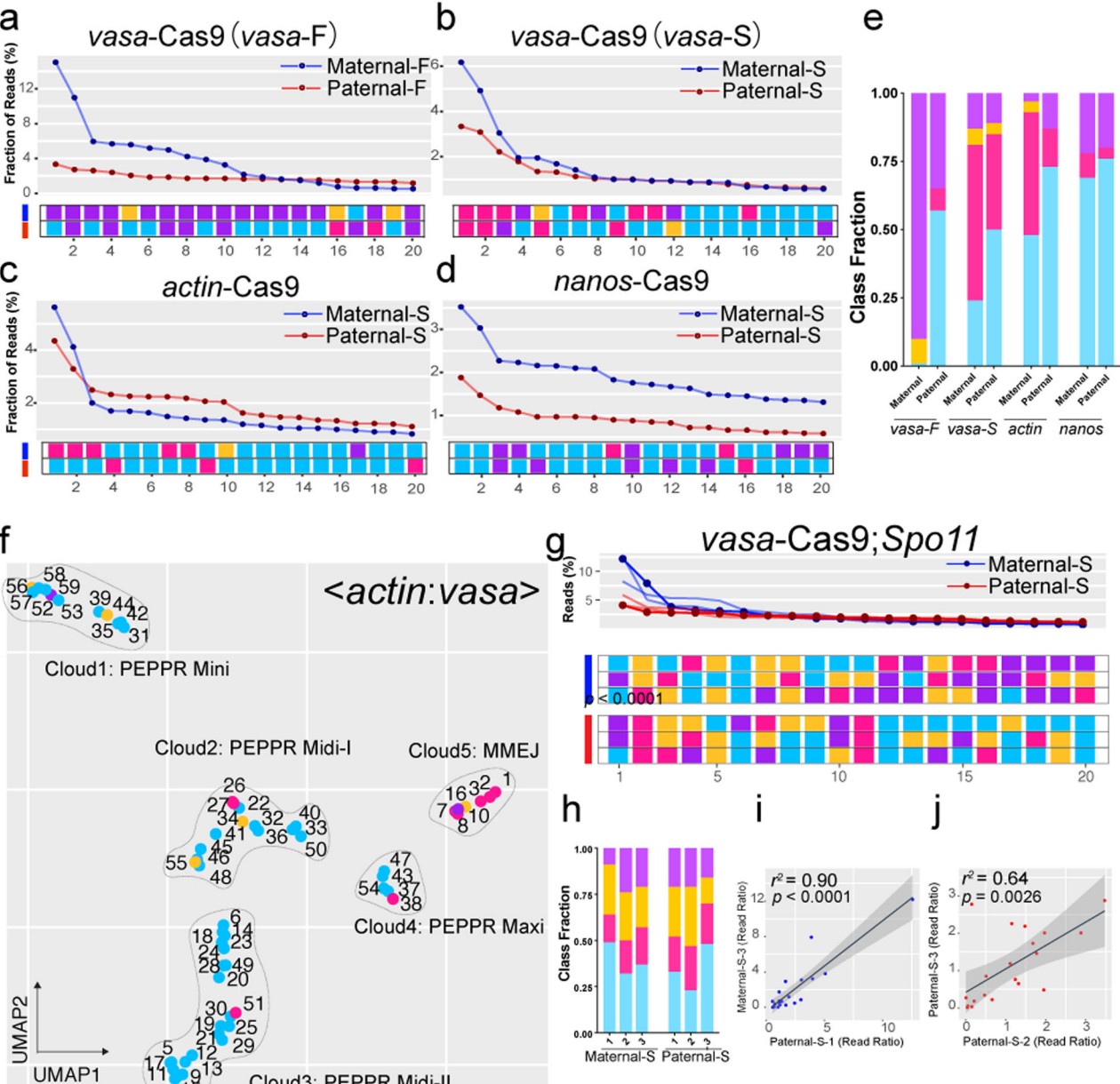

**Fig. 3 | Reproducible distinctive DSB repair fingerprints observed with different Cas9 sources and a second genomic locus. a–d** Unique DSB repair signatures obtained using different Cas9 sources are displayed with the top 20 alleles (landscapes and DSB repair pattern fingerprints). NGS sequencing was performed on pools of 20 adults. **a** *vasa*-Cas9 inserted in the X chromosome and the *ple*CC element carrying the gRNA were both carried by either female or male parents, mimicking a full-drive configuration (Maternal-F and Paternal-F crosses with *vasa*-Cas9). **b** *vasa*-Cas9 split crosses wherein the Cas9 transgene was transmitted either maternally (Maternal-S) or paternally (Paternal-S) and the *ple*CC gRNA bearing cassette was carried by the other parent. Same Maternal-S versus Paternal-S crosses as in panel **b**, but using either *actin*-Cas9 (**c**) or *nanos*-Cas9 (**d**) sources. **e** Class Fraction Index for crosses in panels **a–d**. Bars are shaded according to allelic class color codes. **f** UMAP embedding for visualizing a common set of 59 alleles shared between the four split crosses with *actin*-Cas9 and *vasa*-Cas9. Dots represent single alleles, and the colors indicate the allelic category. **g** Distribution of top 20 alleles generated from single flies derived from a cross between parents carrying the *Spo11* gRNA and *vasa*-Cas9 elements (Paternal-S cross: red lines and Maternal-S cross: dark blue lines). The top plot shows the allelic landscape for the top 20 alleles from all six sequenced single flies and the bottom shows three examples of the classification fingerprints (with all allelic classes condensed into single rows) color coded for the allele categories. **h** Class Fraction Index for *Spo11* gRNA crosses. **i, j** Correlation analysis between two replicates from each cross. Dark blue is Maternal-S and red is for Paternal-S. $r^2$ values and $p$-values are indicated. Source data are provided as a Source Data file.

which we ignore here for simplicity, will require additional future scrutiny. The remainder of these alleles, such as allele 44, could be accounted for variability in the exact Cas9 cleavage site (between the 6th and 7th nucleotides counting from the PAM side), with an extra nucleotide being deleted on the PAM-proximal side of the gRNA cleavage site (Fig. 3f)[43,59,60]. Since both of these outcomes were rare, we hypothesized second-order origins for such outlier alleles further validate the robust nature of our ICP platform in

recognizing core primary categories of DNA repair outcomes. We also analyzed the common 59 alleles by plotting their read frequencies and observed that the differences between the allelic landscapes for the two reciprocal crosses per each Cas9 source mirrored the trend in Fig. 3a–d described above (Supplementary Fig. S2a, b). Cumulatively, these concordant findings support a key role for the parental origin of Cas9 serving as a major determinant of the DSB repair outcome.

## Similar distinctive DSB repair fingerprints are observed at other genomic target sites

Another obvious determinant of DSB repair outcome is the local genomic DNA context. We assessed the general applicability of the ICP by employing it to classify alleles generated by gRNAs targeting four other loci: prosalpha2 (prosα2), Rab11, Spo11 and Rab5 using the vasa-Cas9 source[61]. Paralleling our findings from the ple locus, we observed divergent allelic profiles between Paternal-S and Maternal-S crosses with distinct dominant mutation categories based on the specific target site. For example, the predominant allelic classes generated at the Spo11, prosα2 and Rab11 loci were PEPPR and INSRT alleles, while PEPPR and MMEJ alleles were most prevalent for the Rab5 targets (Fig. 3g, h, Supplementary Figs. S3–6). Among these four targets, Spo11 displayed the greatest divergence in the prevalence of top alleles generated from Maternal-S and Paternal-S crosses (reminiscent of the fine distinctions parsed for the ple locus, Fig. 3g). We nonetheless still observed high correlation coefficients between two replicates within the same cross and significantly lower correlation coefficients associated with inter-cross comparisons between maternal versus paternal Cas9 inheritance (averaged $r^2 = 0.33$, Fig. 3i, j, Supplementary Fig. S3). We also observed distinctive sex-specific DSB repair patterns for Cas9 transmission at the prosα2 and Rab11 gRNAs targeting sites (Supplementary Figs. S4 and S5), although these differences were less pronounced than for ple and Spo11 gRNAs, while for Rab5, the allelic patterns were similar for both maternal and paternal crosses (Supplementary Fig. S6, see Supplementary Discussion Section). In summary, these data support the broad utility of the ICP pipeline to deliver unique discernable locus-specific fingerprints associated with distinct parental inheritance patterns of Cas9 that generalize to other genomic targets.

## Highly divergent maternal versus paternal DSB repair patterns in mosquitoes

Given the strong Cas9 inheritance-dependent distinctions observed for allelic profiles resulting from maternal versus paternal Cas9/gRNA-induced DSBs in Drosophila, we wondered whether similar DSB repair pattern fingerprints could be discerned in mosquitoes carrying a linked "full" gene-drive in which the Cas9 and gRNA transgenes are carried together in a single cassette[62–65]. We examined this possibility using the transgenic An. stephensi Reckh drive, which is inserted into the kynurenine hydroxylase (kh) locus[63]. Because of the Cas9 and gRNA linkage, the Reckh drive behaves as the Maternal-F and Paternal-F cross configurations described above in which all CRISPR components are carried by a single parental sex[63].

Consistent with our observations in flies, the Reckh Maternal-F crosses generated a high proportion of indels that were dominated to a remarkable extent by single mutant alleles with read percentages exceeding 85% for each of the three single mosquitoes sequenced, followed by a long distributed tail of lower frequency alleles. The highly biased nature of the replicate allelic distributions is readily revealed by a virtual step-function in their rank-ordered allelic landscapes (Fig. 4a). In striking contrast, over 50% alleles recovered from the Paternal-F crosses were wild-type (WT), which presumably reflects alleles that either remained uncut or DSB ends that were rejoined accurately without further editing. The highly predominant WT allele was followed by a very shallow tail distribution of low frequency mutant alleles in the paternal rank-ordered allelic landscapes (Fig. 4a). This dramatic difference in allelic profiles between Maternal-F versus Paternal-F crosses was also clearly displayed by the class-tally bars color coded for the different fractions of each class (black = WT) located beneath each landscape (Fig. 4a). Here, the Class Fraction Index measure indicated that Maternal-F crosses generated a greater proportion of INSRT alleles in the first two samples, while Paternal-F crosses produced a high frequency of PEPPR alleles (Fig. 4b). As in the case of allelic profiles recovered at the ple and Spo11 loci in flies,

common sets of highly correlated mutant DSB repair fingerprints were observed across all three replicates of the Paternal-F Reckh crosses (Supplementary Fig. S7). A similar comparison of allelic distributions in the maternal crosses was precluded by virtue of the single highly dominant alleles and corresponding paucity of lower frequency events, the nature of which varied greatly between replicates. We conclude that the high-resolution performance of the ICP platform in Drosophila can be generalized to other insects such as An. stephensi to robustly discern sex-dependent CRISPR transmission patterns resulting in distinct DSB repair outcomes.

## Developmentally regulated DSB repair outcomes

Given the dramatic differences we observed in the frequency and nature of somatic alleles generated in maternal versus paternal-sourced Cas9 in both flies and mosquitoes, we wondered whether the developmental timing of Cas9/gRNA expression (maternal = early? and paternal = late?) was the key determinant for these highly reproducible DSB repair fingerprints. We tested this hypothesis by assessing whether DSB repair fingerprints varied as a function of developmental progression using a series of narrowly timed sample collections of $F_1$ mosquitoes produced from crosses of Reckh parents to WT and assayed DSB repair spectra using the ICP pipeline at 12 different developmental stages (Fig. 4c. Note: as homozygous Reckh transgenic mosquitoes were crossed to WT, all $F_1$ progeny carried one Reckh allele and one WT receiver allele, the latter of which was amplified for DSB repair analysis). We tracked a diminishing proportion of WT (presumably uncut) alleles and a corresponding increase in mutant alleles of various classes at each of the time points (Fig. 4d). Strikingly, nearly half of the target alleles were edited in embryos by 30 minutes post-oviposition for both the Maternal-F and Paternal-F Reckh crosses, which corresponds to early pre-blastoderm stages prior to the maternal-to-zygotic transition, suggesting a very early activity of Cas9 in mosquito embryos driven either by maternally inherited Cas9/gRNA complexes or potentially by very early zygotic expression of the Cas9 and gRNA components (Fig. 4d)[66]. We also observed similarly frequent indels being generated as early as 30 min in flies expressing Cas9 (either maternally or paternally) with a gRNA targeting the prosα2 locus, although the dynamics of Cas9 production are distinct in these two organisms (Supplementary Fig. S8a). Following this initial surge in target cleavage, we observed divergent trajectories in the accumulation of mutant alleles between maternal versus paternal lineages. As an overall trend, mutant alleles accumulated progressively in the Maternal-F lineage until virtually no WT alleles remained, while in Paternal-F lineage, even at the endpoint of adulthood, approximately 60% of WT alleles persisted, in line with our single time point experiments (Fig. 4a, d, Supplementary Fig. S8b). As observed in the final distributions of adult alleles, progeny from Maternal-F crosses tended to be enriched for INSRT alleles over the entire developmental time course, while PEPPR alleles were more common in Paternal-F crosses with pronounced accumulation of such alleles during later stages (Fig. 4e). A finer scale analysis of the categories of mutant alleles generated over time revealed dynamic patterns of prevalent alleles during mosquito developmental stages (Fig. 4e). For example, the proportion of MMEJ alleles peaked at the 2-hour and 4-hour time points (Fig. 4e). Similarly, a split-drive expressing a gRNA targeting the Drosophila prosα2 locus generated distinct temporal profiles of cleavage patterns in crosses from female versus male parents carrying the drive element (Supplementary Fig. S9).

One unexpected feature of the developmental variations in allelic composition we observed was that the proportion of WT alleles increased at certain time points (e.g., 1-hour in maternal cross and 12-hour - day 1 = 24 h in paternal cross). These temporal fluctuations were also observed in flies expressing Cas9 and a prosα2 gRNA at two hours after oviposition (Supplementary Figs. S8a and S9), revealing that this phenomenon might reflect a generally relevant form of clonal

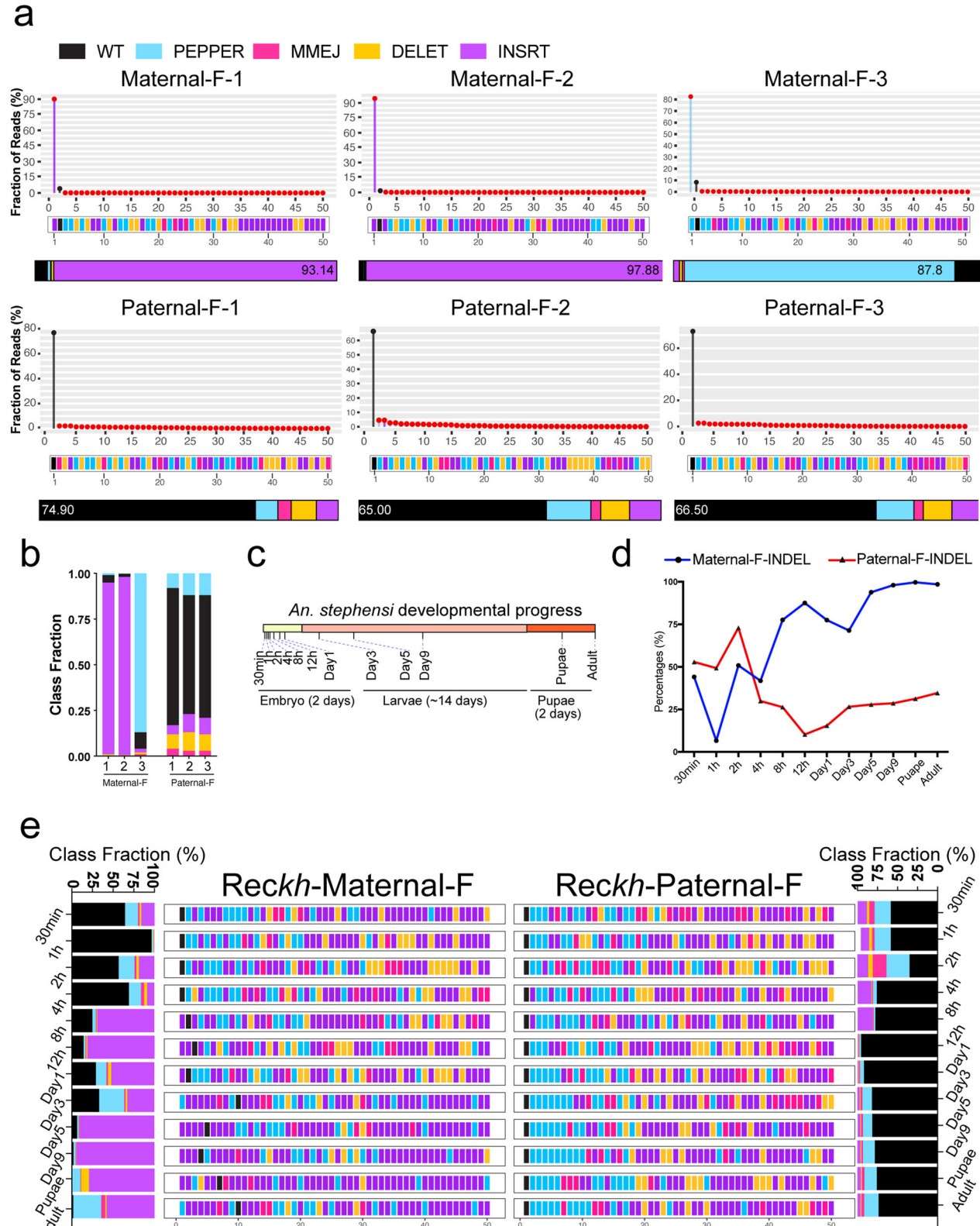

selection for WT cells during pre-blastoderm stages. The latter clonal selection might arise if mutant cells experienced negative selection at certain development stages. In the case of paternal transmission, one strong line of evidence supporting this WT clonal selection hypothesis is that in adults, the Re*ckh* element is transmitted to over 99% of F₁ progeny, indicating that nearly all target alleles in the germline must be WT. This high frequency of paternal germline transmission is also

consistent with the high prevalence of WT alleles tallied at 12 h in embryos derived from the paternal crosses (Fig. 4e, see Supplementary Discussion Section for more in-depth consideration of this point). We analyzed the developmental distributions of 21 common alleles that were generated at all time-points (Supplementary Fig. S10a–e). Most of these common alleles belonged to the PEPPR class, while only five were INSRT alleles, despite the INSRT class overall being the most

**Fig. 4 | Deciphering DSB repair outcomes generated by the *An. stephensi* Rec*kh* drive. a** Rank-ordered landscapes of the top 50 alleles generated from NGS analysis of single mosquitoes. Colored bars with red dots indicate mutated alleles, and black bars with black dots indicate an unmutated WT allele. Middle panels: allelic class fingerprints color coded as in previous figures. Bottom bars: fraction of each allelic class, including WT (black), PEPPR (sky-blue), MMEJ (deep pink), DELET (orange) and INSRT (purple). Numbers indicate the percentage of the corresponding class. **b** Class Fraction Index for single mosquito sequencing data in panel **a**. **c** Developmental time-points for sample collections. **d** Kinetics of Cas9

mutagenesis generated by the Rec*kh* gRNA. Lines represent the summed fraction of mutant alleles at each time-point. Dark-blue lines indicate maternal (Maternal-F) crosses and red lines paternal (Paternal-F) crosses. **e** DSB repair fingerprints at different timepoints. Samples were collected at the time points shown in panel **c** and 20 eggs, larvae, pupae or adults were pooled together for genomic DNA extraction and deep sequencing. The far left and far right panels indicate the Class percentages including WT alleles (black), displaying the proportion of each class at single time-points. Source data are provided as a Source Data file.

prevalent for both crosses, again suggesting that INSRT alleles have a higher diversity than other mutation categories (Supplementary Fig. S10a). Overall, this analysis is in line with our previous observation that Maternal-F crosses produced more INSRT alleles while Paternal-F crosses generated a preponderance of PEPPR alleles (Supplementary Fig. S10b).

## Lineage tracing

Given the strong influence of maternal versus paternal origin of Cas9 on the resulting distributions of alleles characterized above by ICP analysis, we wondered whether such allelic signatures could be exploited for lineage tracing in randomly mating multi-generational population cages. We first examined ICP outputs from a controlled crossing scheme carried out over three generations with *ple*CC and Rec*kh* gRNAs to derive allelic fingerprints distinguishing parents of origin by identifying both somatic alleles in the $F_1$ generation as well as assessment of which of those alleles might be transmitted through the germline to non-fluorescent progeny (i.e., those not inheriting the *ple*CC or Rec*kh* element) at the $F_2$ generation (Fig. 5a–d, Supplementary Fig. S11). As anticipated, in both *ple*CC and Rec*kh* Maternal-F crosses, single dominant somatic alleles were observed in the $F_1$ generation, with the top single allele representing more than 50% of all alleles (Fig. 5a, c). Furthermore, all such predominant somatic mutant alleles, which precluded gene-cassette copying of the *ple*CC or Rec*kh* drive elements in those $F_1$ individuals, were transmitted faithfully through the germline to non-fluorescent $F_2$ progeny with approximately 50% frequency. Furthermore, we observed marked differences in the other half of total reads in $F_2$ progeny depending on the origin of Cas9/gRNA complexes. Thus, a distribution of multiple diverse low frequency mutations were generated when crossing $F_1$ *ple*CC$^+$ or Rec*kh*$^+$ females with WT males (presumably derived from $F_1$ drive females having deposited Cas9/gRNA complexes maternally that then acted on the paternally sourced WT allele somatically in $F_2$ individuals). In the reciprocal male cross, however, approximately 50% of all alleles remained WT (Fig. 5b, d, Supplementary Fig. S12a–f). These findings support the hypothesis that the top somatic indels derived from maternal Cas9 sources were generated at very early developmental stages (possibly at the point of fertilization or shortly thereafter during the first somatic cell division), resulting in a single mutant allele being initially produced and then transmitted to every descendent cell including all germline progenitor cells[49]. With the paternal-sourced Cas9 and gRNA, arrays of variable somatic mutations were recovered with the most prominent alleles accounting for fewer than 10% of the total alleles in $F_1$ progeny (Fig. 5b). Accordingly, paternally generated $F_1$ somatic alleles were more randomly transmitted via the germline of individuals that failed to copy the gene cassette for either the *ple*CC or Rec*kh* elements. As a result of this diversity of somatic $F_1$ alleles, only occasionally were the most prevalent alleles also transmitted through germline (e.g., individuals 1, 4 and 5 in Fig. 5b, Supplementary Fig. S12g–l).

The Rec*kh* element in mosquitoes performed similarly to the fly *ple*CC, however, Rec*kh* $F_1$ individuals displayed less frequent zygotic cleavage and a corresponding reduction in the diversity of resulting somatically generated mutations (>50% WT alleles remained, Paternal-F cross). Consistent with this limited number and array of somatic

mutations in the $F_1$ generation from Paternal-F cross, NHEJ mutations were only rarely transmitted to the $F_2$ generation, probably due to more germline-restricted expression of *vasa*-Cas9 in mosquitoes as compared to flies (Fig. 5c, d). These results again suggest that cleavage and repair events were generated later during development in paternal crosses resulting in a stochastic transmission of $F_1$ somatic alleles to the germline, which were largely uncorrelated with the most prevalent allele present somatically in the $F_1$ parent[49]. Taken together, these highly divergent sex-dependent DSB repair signatures suggested that such genetic fingerprints could be used to track parental history in the context of randomly mating multi-generation population cages.

Based on the highly dominant mutant indels (Maternal-F) versus WT (Paternal-F) alleles generated by Rec*kh* genetic element described above, we evaluated inheritance patterns of indels in multi-generational cages initiated by a 5% introduction of Rec*kh* into WT populations either through maternal or paternal lineages in the $F_0$ generation (Fig. 5e). We randomly selected at least 20 fluorescence marker-positive mosquitoes (10 females and 10 males) for NGS analysis at generations 2 and 3, when the Rec*kh* allele was still present at relatively low frequencies in the population and random mating was more likely to have taken place between Rec*kh*/+ heterozygous and WT mosquitoes. Thus, we envisioned that the source of Rec*kh* allele could be tracked back to a male versus female parent of origin by examining whether a dominant WT allele was present (inherited paternally) or not (inherited maternally) (Fig. 5e, f). Following this reasoning, we inferred a strong bias for progeny inheriting the Rec*kh* element from a Rec*kh*$^+$ males mating with WT females during generations 2 and 3 than the reverse (i.e., female transmission of Rec*kh* alleles) in the maternally seeded lineage. Indeed, in one maternally seeded replicate (cage 2, generation 3), 100% of the progeny had inherited the Rec*kh* element from their fathers (Fig. 5f). In contrast to the striking sex-specific transmission bias observed in maternally seeded cages, progeny from paternally seeded cages displayed more evenly distributed stochastic parental inheritance patterns (Fig. 5f). These highly reproducible parent of origin signatures demonstrate the utility of ICP in allelic lineage tracking, which could be of great potential utility in evaluating alternative initial release strategies for gene-drive mosquitoes as well as post-release surveillance of gene-drives as they spread through wild target populations (see Discussion).

## Marker-free tracking of gene cassettes

Another important challenge for deciphering DSB repair outcomes is to track both NHEJ and gene-cassette mediated HDR events within the same sample. Such a comprehensive genetic detection tool could have broad impactful applications (see Discussion). For example, one important and non-trivial application is to follow the progress of gene-drives in a marker free fashion as they spread through insect populations. Such dual tracking capability would address the potential concern that mutations eliminating a dominant marker for the gene-drive element could evade phenotype-based assessments of the drive process. Accordingly, we devised a three-step short-amplicon based deep sequencing (200–400 bp) strategy based on tightly linked colony-specific nucleotide polymorphisms distinguishing donor versus receiver chromosomes to detect copying of two CopyCatcher elements, *ple*CC and *hth*CC, from their chromosomes of origin (donor

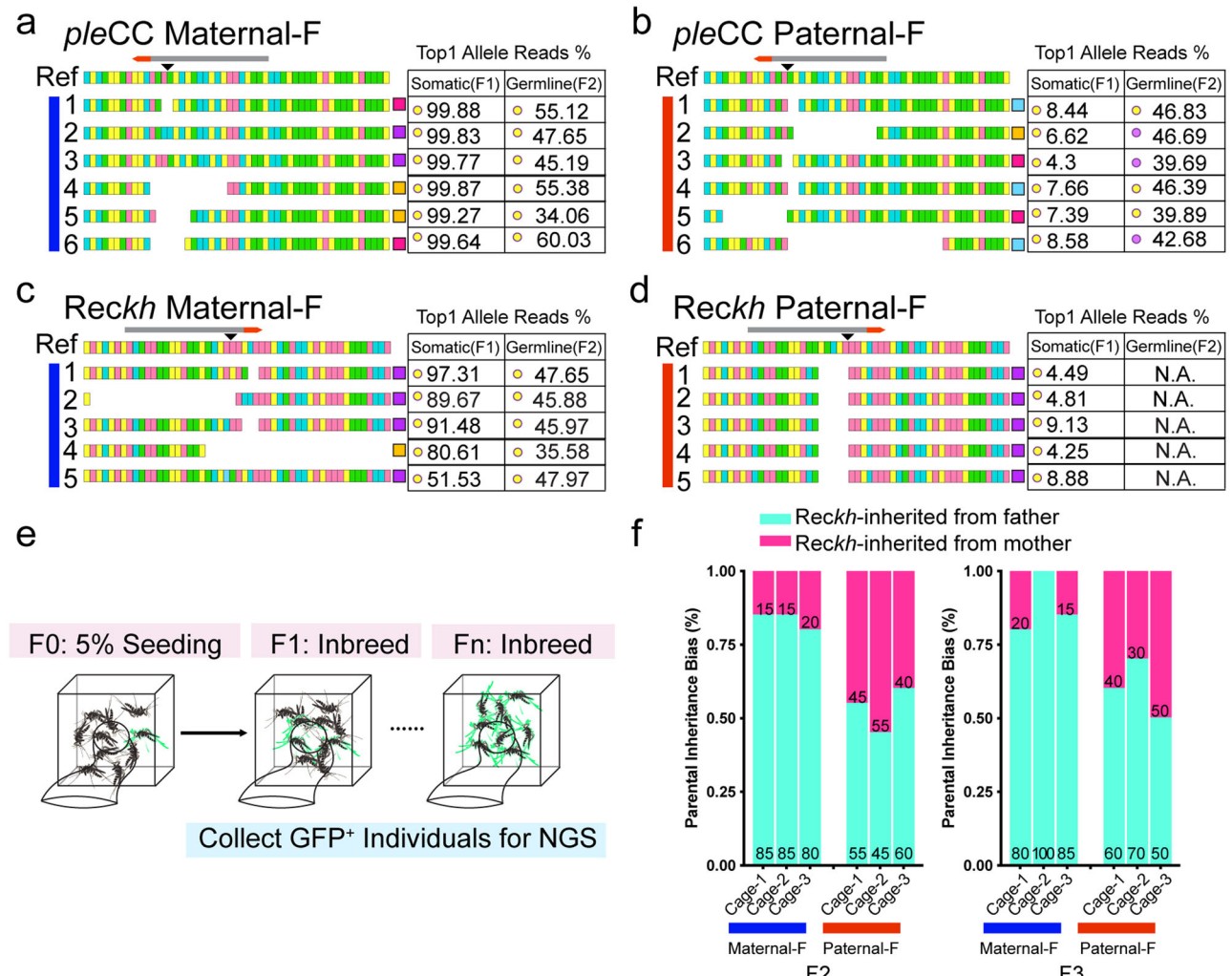

**Fig. 5 | Allelic tracking with the ICP.** Primary DNA sequences of top single alleles and their percentages of the total alleles from six individual sequenced flies derived from *ple* gRNA Maternal-F (**a**) and Paternal-F (**b**) crosses. Gray bars indicate the location of the gRNA protospacer and red arrowheads are the associated PAM sites. The first row depicts the reference sequence covering the expected DSB cleavage site. Colored squares in the right column indicate the class to which a given allele belongs to. The tables shown on the right of each allele show its frequency among all reads. Left columns of the table indicate frequencies of the somatic allele, and the right columns are the top germline mutant allele frequency obtained by sequencing $F_2$ non-fluorescence progeny derived from same $F_1$ individuals whose top somatic allele is displayed in the left column (excluding WT alleles). Colored dots indicate different alleles with the same color shared between two columns indicating that the same allele appeared as both top 1 somatic and germline indels from the same $F_0$ founders. **c**, **d** Allele profiles generated by Re*ckh* parents and progeny generated with the same crossing scheme as for the *ple*CC. **c** Tabulation of the Maternal-F cross. **d** Tabulation of the Paternal-F cross. **e** Crossing scheme for the Re*ckh* cage trials. Three individual cages were seeded with 10 homozygous Re*ckh* females, 90 WT females and 100 WT males for the maternally initiated lineage, while the paternally initiated cages were seeded with 10 homozygous Re*ckh* males, 90 WT males and 100 WT females. At each of the following three generations, 10 Re*ckh*+ females and 10 Re*ckh*+ males were randomly collected for single mosquito deep sequencing. **f** Biased inheritance of Re*ckh* was observed in the maternally seeded cages at generations 2 and 3, but not for the paternally seeded cages. Pink bars denote the fraction of sequenced individual mosquitoes inheriting Re*ckh* from female parents, and cyan colored bars represent Re*ckh* inheritance from the males. Source data are provided as a Source Data file.

chromosome) to WT homologous (receiver chromosome) targets (Fig. 6a)[49]. Notably, this strategy only amplified the inserted gene cassette on the donor chromosome and or the cassette if it copied onto the receiver chromosome. Thus, the measured allelic frequencies indicate the relative proportions of gene cassettes copied to the receiver chromosome versus those residing on the donor chromosome (Fig. 6b displays the inferred somatic HDR frequency quantified from the three-step NGS sequencing protocol as well as Indels quantified by our standard 2-step NGS sequencing protocol - see Methods section for additional details).

In our first set of experiments, we analyzed editing outcomes by examining $F_1$ progeny derived from Maternal-S and Paternal-S *ple*CC crosses. We compared the rates of somatic HDR measured by NGS analysis to those evaluated by image-based phenotypes associated

with copying of the CopyCatcher element. As summarized previously, CopyCatchers such as the *ple*CC are designed to permit quantification of concordant homozygous mutant clonal phenotypes (e.g., pale patches of thoracic cuticle and embedded sectors of colorless bristles), with underlying DsRed+ fluorescent cell phenotypes[49]. Individual flies in which imaging-based analysis had been conducted were then subject to separate NGS HDR-fingerprinting and INDELs-fingerprinting resulting in a comprehensive quantification of HDR, NHEJ, and WT alleles within the same sample (Fig. 6b, libraries for HDR-fingerprinting and INDELs-fingerprinting were prepared from the same individual fly, but with different DNA preparation and sequencing protocols as detailed description in Methods). For these experiments, $F_1$ flies were genotyped and those carrying both Cas9 and *ple*CC gRNA were used for NGS analysis (data shown here are the inferred frequencies of

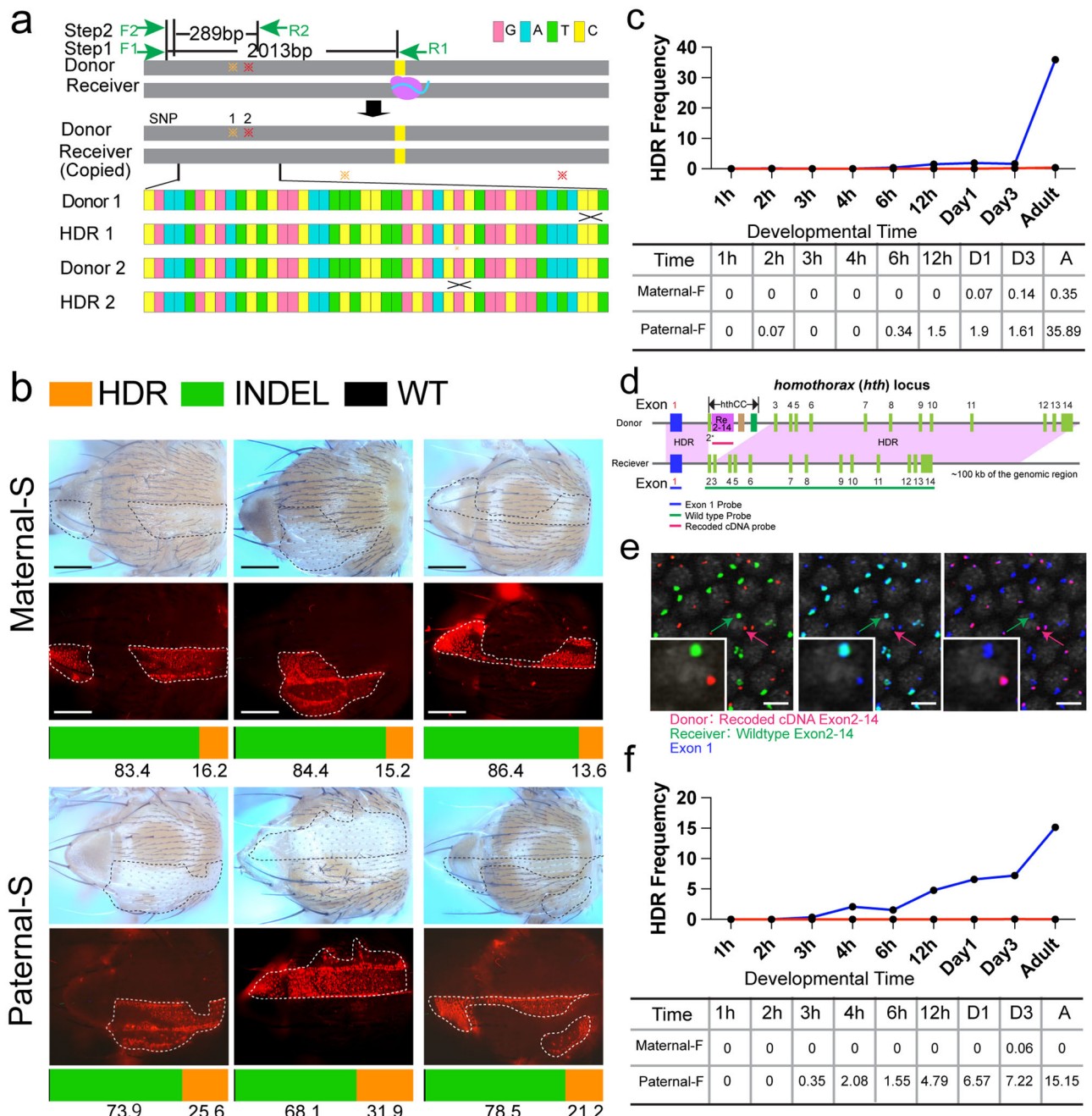

**Fig. 6 | Gene-cassette tracking with ICP. a** Scheme for tracking gene-drive copying using NGS. Gray bars: genomic DNA, pink oval: Cas9 protein, sky-blue line: gRNA, colored asterisks: polymorphisms. Color coded rectangles represent four nucleotides. Four possible recombinants listed are generated by resolving Holliday junctions at different sites marked with black crosses. **b** NGS sequencing-based quantification of somatic HDR generated by *ple*CC in F₁ progeny. Areas delineated by dotted lines indicate patches of cells in which somatic HDR copying events have taken place either under bright field (upper) or RFP fluorescent filed (middle). Bottom bars are the summary of the inferred frequency for the somatic HDR (orange), indels (green) and WT alleles (black) derived from the deep sequencing data using the same samples photographed above. More than three flies from each cross were imaged and used for analysis. Scale bars indicate 200 pixels. **c** Somatic

HDR profile with *ple* gRNA. The red line is for Maternal-F cross and dark blue line for the Paternal-F cross. **d** Diagram of the *hth*CC. Black double arrow: recoded *hth* cDNA, blue rectangles: exon 1, light green rectangles: exons 2-14, and colored lines underneath represent probes used for detection. **e** In situ images with embryos laid from *hth*CC-*vasa*-Cas9 females crossed with WT males. Blue = exon 1, green = WT exons 2-14, red = recoded cDNA for exons 2-14. Insets are magnified single nuclei indicated by colored arrows. This experiment has been repeated at least three times. Scale bars stand for 10 μm. **f** Temporal profiles for somatic HDR-mediated copying of the *hth*CC element assessed by NGS as described for the *ple*CC in panels **c** and **f**. Y-axis tabulates the percentage of HDR at a given time point. Table at the bottom quantifies the HDR fraction at given time points for both the Paternal-F and Maternal-F crosses. Source data are provided as a Source Data file.

---

somatic HDR, NHEJ events, and WT alleles). This dual integrated analysis revealed that HDR in the Maternal-S crosses resulted in ~15% somatic HDR-mediated cassette copying events on average based on sequencing, and that such cassette copying was yet more frequent in Paternal-S crosses, producing ~25% somatic HDR. The nearly two-fold

greater HDR-mediated copying efficiency detected by sequencing in Paternal-S crosses mirrors phenotypic outcomes wherein maternally inherited Cas9 similarly results in a lower frequency of cassette copying detected by fluorescence image analysis in somatic cells than for paternally inherited Cas9 (Fig. 6b)[49].

Our genetic analysis of stage-dependent differences in DSB repair pathway activity in this study is consistent with a commonly held view in the gene-drive field based on a variety of indirect genetic transmission data that HDR-mediated cassette copying does not occur efficiently during early embryonic stages[50,51,63,67–70]. This inference, however, has not yet been verified experimentally. We thus sought to provide direct evidence supporting this key supposition using NGS-based HDR-fingerprinting to track the somatic HDR events across a range of developmental stages in both Maternal-F and Paternal-F crosses in which the Cas9 and gRNA transgenes are transmitted together either maternally or paternally using our validated NGS sequencing protocol. Notably, we collected samples at 9 timepoints and pooled 20 $F_1$ progeny together for pooled sequencing to prime the developmental profile of somatic HDR with *ple*CC (samples were thus collected without genotyping since it is impractical to genotype individual embryos and young larvae). Because of the limitations imposed by embryo pooling we were unable to use the same samples collected here for also quantifying the generation of somatic NHEJ alleles (i.e., only half of the $F_1$ progeny carried the *vasa*-Cas9 transgene on the X chromosome and those embryos lacking this transgene were not suitable for generating mutations - note that such an analysis was possible in the case of the viable Re*ckh* drive shown in Fig. 4e as well as for a viable split-drive allele inserted into the essential *prosalpha2* locus shown in Supplementary Fig. S9). Indeed, NGS analysis detected only very rare examples of somatic HDR events in early embryos derived from both crosses (Fig. 6c). Notably, HDR in the Paternal-F cross detected by this sequencing protocol increased substantially to 35.9% during adult stages, a period coinciding with the temporal peak of the *pale* expression profile (note that in this experiment we employed the *actin*-Cas9 rather than *vasa*-Cas9 source, which has higher level of Cas9 expression in somatic cells and generates a correspondingly higher frequency of somatic HDR)[49].

We extended our sequencing-based strategy to quantify somatic HDR using a second CopyCatcher element (*hth*CC) designed specifically to identify even rare copying events in early blastoderm-stage embryos. The *hth*CC is inserted into the *homothorax* (*hth*) gene and was engineered to visualize HDR-mediated copying of the gene cassette by fluorescence in situ hybridization (FISH) using discriminating fluorescent RNA probes complementary to specific endogenous versus recoded cDNA sequences (Fig. 6d, e). In this system, copying of the transgene from the donor chromosome to the receiver chromosome would be indicated by the presence of two nuclear dots of red fluorescence detected by the *hth* recoded cDNA-specific probe (indicating two copies of recoded *hth* cDNA). In contrast, cells in which no copying occurred should contain only a single nuclear red dot signal (from the donor allele). Such in situ analysis detected no clear case of gene cassette copying in any of the ~5000 blastoderm stage cells examined across ~500 embryos (with the caveat that some mitotic nuclei generate ambiguous signals depending on their orientation). This qualified negative result assessed by in situ analysis was consistent with the very low estimates of HDR frequency during the same early blastoderm-stage developmental window based on NGS analysis in staged time-course experiments, although the latter sequencing method did detect very low levels of somatic HDR at ~3 hours after egg laying from the Paternal-F crosses (and no copying until day three of larvae with the maternal cross – Fig. 6d–f). The very low levels of somatic HDR observed in early embryos for the *hth*CC construct either by in situ hybridization or by NGS sequencing parallel the results summarized above for the *ple*CC element (Fig. 6c, f). The maximal somatic HDR frequency observed for the *hth*CC Maternal-F crosses (0.06% at day 3 after egg laying) was somewhat lower than that for the similar cross for *ple*CC (0.35% at adult stage), consistent with the predominance of single mutant alleles being generated at very early stages following fertilization in Maternal-F crosses. In contrast to the exceedingly rare copying of the *hth*CC element detected in early

embryos for either the Maternal-F or Paternal-F crosses, the same element frequently copied to the homologous chromosome during later developmental stages in Paternal-F crosses as assessed by NGS sequencing. The *hth*CC element again copied with somewhat lower efficiency than the *ple*CC element (e.g., 15.2% for *hth*CC versus 35.9% for *ple*CC tabulated in adults), presumably reflecting differing genomic cleavage rates or gene conversion efficiencies generated by their respective gRNAs (including total cleavage levels and temporal features). In aggregate, these two examples of quantitative analysis of copying frequencies based on both NGS and in situ analysis demonstrate that ICP and NGS-based quantification of gene conversion events can be successfully integrated for a comprehensive analysis of DSB repair outcomes, including both NHEJ and HDR events as a function of developmental stage. These powerful tools also could be applied for following gene-drive spread through freely mating populations in a marker-free manner as well as for a variety of other applications including gene therapy (see Discussion).

## Discussion

### The ICP generates broadly applicable robust and discriminating DSB repair signatures

A key advantage of the discriminating and highly informative DSB repair signatures generated by the ICP is the ability to track combinations of genetic lesions and gene-editing events in complex tissues composed of diverse cell types. In this study, we provide several proof-of-principle demonstrations of the utility of the ICP including discovery of a robust developmental progression in DSB repair pathway choice, the ability to track parent of origin for gene-drive systems - including the challenging scenario of freely mating individuals in multigenerational population cages, and marker-free quantification of both specific mutations and interhomolog copying of a gene cassette in the same sample.

In comparison to prior sequence analysis pipelines such as those elegantly developed by Hussmann and other groups[9], our ICP offers the following advantages: 1) the ICP platform can be flexibly adapted to different endogenous genomic loci and can be employed in complex developing multicellular organisms in a non-invasive manner; 2) virtually all reads can be analyzed as long as the dictionary includes all possible repair outcomes, 3) by using a 24-nt seed region for allele fishing and classification the ICP platform is more straightforward to use and readily identifies the vast majority of technical-based (e.g., PCR or sequencing) errors while at the same time overcoming issues arising from segment alignment-based classification methods[9], and 4) ICP outputs intuitively rank-ordered and color-coded fingerprints that reproducibly identify the most frequent allelic category profiles and overall DSB repair patterns when compared across different experimental settings (e.g., diverse inheritance patterns of different CRISPR components or across developmental stages).

### Discovery of a developmental progression in DSB repair choice

It is well appreciated that various types of mutant alleles can be generated in response to repair of DSBs in different cellular contexts and at different genetic loci[61]. In *Drosophila*, significantly different editing outcomes have also been observed based on maternal versus paternal inheritance of CRISPR components[49,50,61]. Here, we substantially extend these findings using highly discriminating ICP analysis discovering a robust developmental progression of DSB repair pathway choice. In early blastoderm embryos of both fruit flies and mosquitoes we find a stereotyped sequence of repair pathway usage in which the earliest repair events tend to be mediated by the MMEJ pathway, followed by a distinct subset of NHEJ alleles (e.g., INSRT in maternal crosses, PEPPR in paternal crosses), and then only later (post-blastoderm/adult) by efficient HDR.

One interesting trend in these studies was the prevalence of MMEJ repair during early embryonic stages from maternal crosses, which is

consistent with the importance of MMEJ as a primary DSB repair pathway during mitosis since the rapid cell-cycles occurring in pre-blastoderm embryos are composed almost entirely of short S and M phases[71–74]. Similarly, a predominance of MMEJ events was noted in analysis of mutations generated by population suppression gene-drive systems in *An. gambiae*[75]. Future studies employing RNAi or CRISPRi to silence expression of factors required specifically for MMEJ versus other branches of DSB repair may shed further light on this interesting association[9,76,77]. A more general role of cell-cycle phase might be another fruitful avenue to investigate, since prolonged association of Cas9/gRNA complexes with DNA targets, as is likely to take place in paternal crosses, may result in the preferential generation of PEPPR alleles we observed or MMEJ events as has been reported in zebrafish embryos[78].

The ICP could also be combined with other existing bioinformatic tools to meet challenges broadly facing current approaches. Thus, the ICP could be integrated with various existing next-generation sequencing (NGS) tools that enable scalable detection and quantification of targeted mutagenesis and comprehensive marker-free investigation of genome editing efficacy and specificity, which remains a great challenge for unambiguous and in-depth decomposition of the diverse DNA lesions[9,33]. These existing mutant analytic pipelines are highly dependent on the local alignment or position of edited nucleotides and often do not account for the a priori nature of target sequences, which weakens the underlying link between DSB outcomes and operative repair pathways[33,79]. Furthermore, nearly all current DSB classifier systems assess DNA repair events in homogeneous cell types such as cultured cell lines, leaving unresolved how diverse cell fates or alternative potential emphasis of repair pathway choice during development may influence editing outcomes. With these limitations in mind, the ICP platform could help address many of these challenges by rapid, semi-automatic at error-calling, and adaptable resolution of complex mutations that are processed and distilled into informative color-coded graphical outputs of ranked mutation classes. In principle, these advantages should also be applicable to intact vertebrate organisms, for example to aid the characterization and parsing of various off target effects of gene editing in human cells that may take place in diverse tissues in response to gene therapy interventions.

### Tracking parent of origin for gene-drive transmission
Analysis of DSB repair distributions generated from six genomic targets and eight different genetic crossing schemes revealed highly distinctive ICP fingerprints resulting from maternal versus paternal transmission of Cas9 in both flies and mosquitoes. These trends were robustly revealed both by analysis of highly predominant alleles and by overall prevalence of those allelic classes among the top alleles. For example, regarding gene-drives, surveillance of specific gene edits (indels or gene-cassette) can serve as robust identifiers of maternal versus paternal inheritance of a specific indel or gene-drive element. Thus, in maternal crosses we observed highly prevalent single mutant alleles and no remaining wild-type alleles. Such dominant maternally generated alleles were then transmitted to nearly all progeny. In contrast, paternal Reckh transmission resulted in a large proportion of unmutated wild-type alleles and a broader range of alleles probably due to delayed DNA cleavage and repair. These dynamic and distinct DSB repair signatures should permit inference of the parental sex of an individual insect collected during early phases of a gene-drive release as they did in our laboratory experiments, and could prove invaluable in monitoring and evaluating the spread of a gene-drive element following potential releases into wild populations, as well as management and follow-up analysis of gene-drive performance in such field trials. For example, in population cages, ICP analysis revealed that initiation of drive through females led to a strong subsequent bias in the first few generations in favor of transmitting the gene-drive elements

paternally, while initiation of drive using males resulted in no obvious subsequent sex bias in transmission. One potential explanation for these notably divergent outcomes is that multi-generational accumulation of maternal Cas9/gRNA complexes deposited into eggs by females might decrease the fertility of their daughters, a phenomenon that should not arise in the case of paternal seeding[80]. These and other paradigms for initial release of gene-drives merit further exploration using the ICP platform and could inform decisions regarding what sexes to release in potential field applications (e.g., males only, females only, or combined male/female releases, Fig. 5f).

### Marker-free tracking of gene cassette copying
Our proof-of-principle for deep sequencing-based analysis of HDR-mediated cassette copying demonstrated that ICP also can be integrated with NGS-based sequencing of specific chromosome homologs to track copying of gene cassettes in a marker-free manner. This NGS-based quantitative measurement of cassette copying in somatic cells is highly concordant with our prior phenotypic quantifiable measures assessed with the *ple*CC CopyCatcher element in adults (this study and Li et al)[49], as well as for the *hth*CC CopyCatcher, which we designed to visualize potential copying events in early blastoderm stage embryos (this study). Indeed, the data presented here provide the first direct experimental evidence in support of the hypothesis that DSBs are only very rarely repaired by HDR during the early rapid cell divisions in blastoderm stage embryos[81–84]. The ability to integrate analysis of DSB repair outcomes including both NHEJ and gene-conversion outcomes in tissues comprised of complex cell types provides a powerful tool for comprehensive analysis of DSB repair mechanisms in diverse multicellular contexts and should provide practical guidance for how best to manipulate and optimize genetic editors for desired HDR editing.

Integrated ICP and NGS sequence analysis also provided a proof-of-principle for tracking gene-drive elements in a marker free and non-invasive fashion, which should be of considerable value to aid monitoring of future potential field implementations of non-fluorescence marked gene-drive elements (should fluorescence markers incur associated fitness costs). In addition, sequencing-based approaches permit temporal analysis of dynamic HDR profiles during early embryonic as well as later stages of development to precisely pinpoint when such gene conversion events take place. The vital information provided by such high-resolution sequencing tools will inform future design and optimization of diverse gene editing systems.

### Perspectives for future potential ICP implementations
Beyond its varied and highly impactful applications to the gene-drive field, we envision that the ICP platform also could be applied to a broad range of other gene editing contexts in which tracking both accurate editing and off-target mutations are important in intact organisms with complex tissues comprised of multiple different cell types. Such integrated sequence analysis could be employed for lineage tracing, in particular for cancer cell progression. Thus, ICP analysis could be coupled with highly informative single-cell CRISPR/Cas9 based cancer cell lineage tracing strategies, to parse the process of tumor metastasis with yet greater resolution[85–87]. For example, a significant concern with many CRISPR-based gene therapies is the generation of undesired and potentially adverse off-target effects. The ICP platforms could be coupled with other strategies to quantify and characterize such off-target effects by combining it with genome-wide detection methods such as DISCOVER-Seq and CIRCLE-seq to first identify relevant low frequency off-target sites[41,88,89]. Similarly, ICP analysis could potentially contribute to defining and assessing categories of events occurring at candidate mutational hotspots in certain genetic conditions (e.g., fragile chromosome syndromes) or primary versus developing tumors by performing CHIP-seq by using antibodies against to DSB repair core factors (e.g., MRE11). Such an analysis might identify signature recurrent mutations such as NHEJs bordering genome rearrangements due

to cleavage and inaccurate rejoining of broken ends from two different chromosomes (or inversions within the same chromosome). Allelic dictionaries could thus be constructed to follow the occurrence and nature of such relevant recurrent alleles generated during dynamic cancer progression at single-allele resolution by taking tissue biopsies at different stages of tumor progression which may reveal stage specific repair programs during tumor progression. In a similar vein, it should also be possible to adapt the ICP for lineage tracing using endogenous genome targets rather than being restricted to incorporating synthetic DNA recorders into the host genome. Such a non-invasive diagnostic strategy should have broader and more flexible applications compared to most of the currently used recorder systems associated with synthetic barcodes[90,91]. These overall advantages of ICP analysis fulfill the requirements of high-diversity and trackability as an ideal molecular recorder and should be invaluable for in-depth retrospective tracing of the origin of somatic mutations that arise during normal development (e.g., due to failures in DNA repair) or to pathogenic scenarios such as tumor metastasis[86] and chromothripsis[92,93].

More generally, the ability to track both NHEJ and gene-conversion outcomes provides a powerful tool for comprehensive analysis of DSB repair mechanisms in diverse complex multicellular contexts. This dual tracking capability could help address a major concern for the gene therapy field in identifying and tracking bystander mutations within or adjacent to the desired targets during the treatment process[94]. Many efforts aimed to bias HDR editing outcomes have focused on either suppressing activities of NHEJ components or enhancing HDR pathways by tethering the core factors to DSBs[95–98]. ICP-based tracking of these various outcomes should shed light on the role of the genomic DNA context of targeted sequences on repair outcomes in specific organs or complex tissues, perhaps providing guidance for customized regulation of DSB repair pathway activity via adjunctive therapies to suppress the activities of dominant error-prone repair pathways, while promoting desired HDR-mediated edits[94].

In-depth ICP analysis should also be beneficial in the context of detecting rare off-target mutations or genome rearrangements that could present serious health risks accompanying gene therapy. In particular, such a simultaneous analysis would be invaluable in monitoring outcomes of in vivo gene therapy treatments in humans where a diversity of edits might be expected in different tissues, which is a widely appreciated concern[88,89,99,100].

## Limitations of the study

Despite the substantial advances provided by the ICP platform coupled with NGS-based detection of gene-conversion events reported in this study, there are several limitations of the current system. For example, a more accurate and precise definition for classifying the complex alleles would extend the resolution of the platform. In the case of alleles repaired by deletion and insertion, a fraction of such alleles may undergo microhomology mediated deletion and synthesis-dependent insertion[57,58]. Parsing such multiple rounds of editing may increase the resolution of mutational allele categories and should provide a better understanding of the DSB repair mechanisms. Our PCR-based deep sequencing analysis is currently limited to detect indels within a few hundred base pairs of the Cas9 cleavage point. Thus, large deletions, large insertions or rare editing outcomes like chromosome translocations are not currently recovered in our analysis. Future combinational analyses incorporating alternative sequence analysis strategies should help deepen our understanding of Cas9 generated DSB repair outcomes. Also, our preliminary dissection of additional potential DSB repair classes gleaned from UMAP analysis suggested DSB repair mechanisms might be more complicated, possibly reflecting multiple rounds of repair, a potential phenomenon meriting further analysis. Additionally, an algorithm such as that developed by Chen and colleagues could potentially be employed to

automatically build MMEJ dictionaries with no required user input[101]. Similarly, bioinformatic features of the system deployed by Hussmann and colleagues might extend the depth and discrimination to mutant allelic categories based on mechanistic insights into consequences of shifting the DNA repair decision hierarchy in different directions[9]. Integration of such features into future versions of the ICP platform should yet broaden its considerable current utility.

A limitation of our developmental studies was that this analysis differed from that of our single fly or mosquito sequencing in that we pooled DNA extracted from multiple individuals since it is currently technically challenging to use a single embryo to prepare NGS libraries. Such pooling of individuals dilutes inter-individual sequence differences by averaging, and therefore reduces its resolution relative to that obtained from single animal sequencing data. Also, in this experimental design Cas9-dependent editing was cumulative over time, which did not permit an exclusive sampling of specific editing outcomes within narrow temporal windows. We utilized the flexible genetic tools in *Drosophila* such as a heat-shock inducible Gal4 to activate Cas9 expression and then assay the temporal pattern of DSB repair. However, it takes approximately an hour to activate Cas9 expression using this indirect method, which also is associated with significant variation. In future studies such limitations might be overcame by using more direct rapid heat or chemical inducible-Cas9 sources[102–104], to provide sharper temporal peaks of Cas9 activity.

## Summary

In summary, in-depth analysis of DSB repair using the ICP platform has broad future applications to interpretation of DSB repair outcomes permitting tracking of specific mutant alleles as well as copying of gene cassettes. This highly flexible platform and its future refinements offer great promise in analysis of laboratory experiments as well as in providing a new avenue for practical assessment and management of gene editing in efforts for precise gene therapy, as well as genetic manipulation on disease vectors and agricultural pests in various contexts including potential field tests of gene-drive systems.

## Methods

### Animal stocks and genetics

Experimental flies were fed with standard *Drosophila* food under 25 °C with a 12/12 h day/night cycle. *An. stephensi* Reckh drive was maintained in the ACL-2 insectary facilities located in University of California, San Diego, under the condition with 27 °C and 77% humidity. Mosquito larvae were grown with TetraMen fish food (Tetra, #77104-12) mixed with 50% yeast powder (Red Star, #B005KR0MZG), and adults were provided with 10% (wt/vol) sucrose solution. Five days after mating, mosquitoes were fed on defibrinated calf blood (Colorado Serum Co., Denver) using the standard Hemoteck membrane feeding system[63].

gRNAs used in this study were previously reported as components of gene-drive systems, although they were used primarily to detect the somatic rather than germline indels in F$_1$ progeny in the current study. We applied four different crossing schemes including two split-drive crosses (Cas9 and gRNA were separately inherited from parents, Maternal-S: Cas9 provided by females and gRNA by males, Paternal-S: Cas9 provided by males and gRNA by females), and two full-drive crosses (Cas9 and gRNA inherited together from single parent, Paternal-F: Cas9 and gRNA inherited together from males, Maternal-F: Cas9 and gRNA inherited together from females) to mimic the spatial and temporal Cas9 expression levels in flies. The split-drive crosses were performed by crossing flies carrying gRNA inserted at the genomic site targeted by the gRNA and static Cas9 cassettes were inserted elsewhere in the genome. Transgenic Cas9 lines used for split crosses including *actin*-Cas9, *vasa*-Cas9 and *nanos*-Cas9 inserted in *yellow* locus on the X chromosome have been described previously. For full-drive crosses, both Cas9 and gRNA were inserted into the genome at

the gRNA cleavage site and were inherited or copied together. All the protocols used in this study followed procedures and protocols approved by the Institutional Biosafety Committee from the University of California San Diego, complying with all relevant ethical regulations for animal testing and research (protocol #S18147).

### gRNAs

Six different *Drosophila* genomic DNA-targeted gRNAs (*ple*CC: *pale* gene, CG10118, *Rab5*: CG3664, *Rab11*: CG5771, *prosalpha2* (*prosα2*): CG5266, *Spo11*: CG7753, *hth*: CG17117), and 1 *Anopheles stephensi* genome DNA-targeted gRNA (*kynurenine hydroxylase*, *kh*, ASTE004879) under the control of U6 promoter were used in this study (gRNA targeting sequences were listed in Supplementary Table 1). The gRNAs used in this study targeted exons (*Rab5*, *prosα2*, *Rab11*, *Spo11*, *hth* and *kh*) or introns (*ple*CC: *pale* intron 1) of genes essential for viability (*ple*, *Rab5*, *Rab11*, *prosα2* and *hth* are recessive lethal) or reproduction (*Spo11*, which is encoded by *mei-W68*, is recessive sterile). All these gRNAs were stably inserted into genomic DNA and persistently expressed, while Cas9 was provided separately (Rec*kh* is a full drive in which both the Cas9 and gRNA elements are inserted together as a unit into the *kh* locus at the site of gRNA cleavage).

### Time-course assay

Time-course assay was performed with homozygous flies carrying *prosα2* gRNA and X-chromosome sourced *vasa*-Cas9, or homozygous Rec*kh* full drive mosquitoes. DSB repair outcomes were assessed by performing Maternal-F and Paternal-F crosses. For setting up crosses with the mosquito Rec*kh* line, we collected pupae and separated them into female and male cohorts, which were then mated with WT for five days and fed with calf cold blood (Colorado Serum Co., Denver). Two days after blood feeding, mosquitoes were subjected for forced egg laying for 30 min and samples were collected at 12 collection time-points after egg laying including 30 min, 1 h, 2 h, 4 h, 8 h, 12 h, day1, day3, day5, day9, pupae and adults. Cas9 mutagenesis efficiency was calculated by the proportion of each allele relative to the total reads. Allelic frequencies of indels and the WT allele were used for plotting the data.

### Lineage tracking assay

Three generations crosses were performed with *ple*CC and Rec*kh* with maternal versus paternal crosses (Maternal-F and Paternal-F), to determine how the somatic indels were selected and passed through germline cells. For *ple*CC, we combined X-chromosome sourced *vasa*-Cas9 with *ple*CC (inserted in the first intron of the *pale* gene on the third chromosome) to make homozygous stock, and then performed the Maternal-F and Paternal-F crosses. Of note, the homozygous *pale* gene mutation is embryonic lethal, so the third chromosome was balanced with TM6 balancer. At least three replicates were conducted at the same time. With the Maternal-F crosses, we were able to use both trans-heterozygous $F_1$ females and males for outbreeding with the WT, to generate the $F_2$ progeny for examining germline indels. All $F_1$ trans-heterozygous progeny carrying both Cas9 and gRNA were collected for somatic indels sequencing after mating and egg laying, and $F_2$ animals without fluorescence were used for germline indels sequencing.

Rec*kh* cage trials were seeded with 5% of homozygous transgenic mosquitoes with three replicates. In brief, the female lineages were set up with 10 homozygous Rec*kh* females with 90 WT females and 100 WT males, while male lineage was seeded with 10 homozygous Rec*kh* males, 90 WT males and 100 WT females. At each generation, we randomly selected 10 Rec*kh*+ females and 10 Rec*kh*+ males for single mosquito deep sequencing.

### Target amplification and Illumina based deep sequencing

Genome DNA was extracted from twenty embryos, larvae and adults for pooled NGS sequencing, with DNeasy Blood & Tissue Kits

according to the manufacturer (Qiagen, #69504), and followed by column (Qiagen, #69504) purification. Single fly or mosquito genomic DNA was extracted with single fly preparation (crushed with 49 μl lysis buffer: 1 mM EDTA, 10 mM Tris pH 8.2 and 25 mM NaCl, and 1 μl Proteinase K), followed by incubation at 37 °C for 30 min and 95 °C for 2 min.

About 300 ng genomic DNA was used as the template for PCR amplification, with gene-specific primers containing Illumina compatible adapters (Forward: 5'-ACACTCTTTCCCTACACGACGCTCTTCC GATCT-3' and reverse: 5'- GACTGGAGTTCAGACGTGTGCTCTTCCG ATCT-3') at the 5 terminals. Gene-specific primers were designed by subjecting for whole genome blast to get rid of non-specific amplification. A two-steps PCR-based strategy was applied for NGS library preparation. The first round of PCR was performed with gene-specific primers and genomic DNA as templates for 25 cycles amplification. PCR products were verified with gel electrophoresis and subjected for gel purification for extra primers filtration. Purified first-round PCR products were then used as templates for another 5 cycles of PCR, with barcode-containing xGen UDI Primer pairs (IDT, #10005922). Amplicons with distinct index were multiplexed at 10 nM per sample to a final 20 μl volume for Illumina sequencing using Novaseq platform. All primers used in this study were listed in Supplementary Table 2.

Deep sequencing was performed with IGM (Institute of Genomic Medicine, University of California, San Diego). Generated raw reads were demultiplexing using the Barcode Splitter Script by IGM, and then analyzed with the ICP classifier.

### Somatic HDR quantification

Two constructs, *ple*CC and *hth*CC were used for quantifying somatic HDR frequency with deep sequencing, by adapting the DNA library preparation protocol into a three steps polymorphism-based strategy. Firstly, we identified several stable polymorphisms between the donor (*ple*CC or *hth*CC inserted chromosome) and receiver chromosome (WT chromosome inherited from the mated WT parents), for distinguishing the alleles carrying the gene cassette that were amplified from either the donor chromosome or HDR converted receiver chromosome. The first-round amplification was performed with forward primer located beyond the polymorphisms, and reverse primer sitting within the insertion cassette as we illustrated in Fig. 6a, so both the donor chromosome and HDR converted receiver chromosome were successfully amplified (all alleles carrying gene cassette insertion). Secondly, all the first-round PCR products were used as templates for a nested PCR, with a forward primer still beyond the polymorphism but reverse primer near to the polymorphism, producing a 200–400 bp short amplicon that accommodating the NGS protocol. All the following steps were the same to standard NGS library preparation. For quantifying both somatic HDR and Indels in the same sample, NGS libraries were prepared separately by using either our standard two-step short amplicon based PCR (Indels and WT) or the modified and extended three-step polymorphism based PCR (HDR) scheme described above and diagramed in Fig. 5a. The result of somatic HDR sequencing analysis is an estimate of the ratio of gene cassettes located on the receiver chromosomes versus total donor chromosomes (50% of all homologous chromosomes) which can be converted into the fraction of somatic HDR (fraction of HDR-converted receiver chromosome in total receiver chromosome) by the equation: (% Receiver reads/% Donor reads) * 100. While somatic Indels sequencing analysis provides an estimate for the ratio of WT to cut (and variously mutated) receiver chromosomes.

### Quantification and classification of indels

Sequencing reads with the same index were grouped as from the same sample after demultiplexing. Two complementary DSB classifiers, the Nucleotide Position Classifier (NPClassifier) and Single Allele-resolution Classifier (SAClassifier), were built based on ShortRead

package in R. NPClassifier categorized indels according to the start and end position of nucleotides being edited. In brief, alleles with PAM-distal end deletion before the Cas9 cleavage site were assigned into PEPPR class (PAM-End Proximal Protected Repair), any alleles with microhomology-based deletion (annealing of ≥2 nt microhomology sequences and deletion the interval sequences) were assigned to MMEJ class, alleles with deletions excluded from PEPPR and MMEJ were assigned into DELET (deletion), all alleles including insertion without deletion, deletion plus insertion (even 1 nucleotide substitution near to cut site) were categorized as INSRT (insertion).

For making SAClassifier, we firstly created full-length dictionaries for PEPPR, MMEJ and DELET with perfect matched alleles. PEPPR dictionary was synthetically built by iteratively increasing the length of deletions by a single nucleotide distal to the PAM site (excluding alleles belonging to the MMEJ category), with predictable library capacity being defined by the gRNA target site and length of reads (100 bp in this study). MMEJ and DELET full-length dictionaries were built by enumerating a collection of all MMEJ alleles across all samples with NPClassifier outputs. We manually corrected alleles with obvious primer errors (i.e., errors within the target-specific primer binding region) and technical errors (including PCR and sequencing errors, which occur randomly relative to the cutting site) in MMEJ and PEPPR dictionaries, to create perfectly aligned and non-redundant full-length dictionaries. To automatically call for errors, we built three 24-nt short dictionaries which were derived from the full-length dictionaries and contained the seed region spanning the Cas9 cleavage site, since our observation proved most errors located more than 12 nt away from the cut site. With this strategy, these short dictionaries permitted automatic assignment of reads also containing errors to the correctly matched root allele with the same seed region. Regarding the highly diverse nature of the insertion group, we assigned all alleles remaining after initial "fishing" with three major dictionaries (PEPPR, MMEJ and DELET) to the INSRT class.

### Tracking HDR at early embryo stages using fluorescent in situ hybridization (FISH)

To test somatic HDR at early embryo stages, we crossed *hth*CC-*vasa*-Cas9/TM6 females with *Oregon-R* WT males for three days and collected eggs at two hours after oviposition for fluorescent in situ hybridization using the method developed by Kosman[105]. Probes used for this experiment were prepared by either initially amplifying from genomic DNA of *D. melanogaster* (for exon1, intron 1 and WT cDNA) or *hth*CC plasmid (recoded cDNA probe). Sequence validated plasmids expressing each probe under T7 promoter were linearized, purified with phenol/chloroform and then being used for in vitro probe synthesize with hapten-U NTP mix (Perkin Elmer, #NEL 555. Exon1: Dig 488, wild type cDNA: 647 FITC, and recoded cDNA: Bio555). Fragmented probes were used for detecting each transcript in fixed embryos. In situ hybridization process was conducted according to our previous protocol[105]. Embryos were then incubated with primary antibodies of sheep anti-Dig 488 (Roche, #1333089), mouse anti-Bio555 (Roche, #1297597) and rabbit anti-FITC (Molecular Probes, #A-889) respectively at 4 °C overnight (1:100). On the following day, embryos were washed with PBT (PBS with 0.1 v/v Tween 20. PBS, Thermo Fisher Scitific, AM9625. Tween 20, Millipore Sigma, #9005645) three times, and then incubated with secondary antibodies (1:300, Thermo Fisher Scientific, Alexa 488 Donkey anti-Sheep #710369, Alexa 555 Goat anti-Mouse #A-21422, Alexa 647 Chicken anti-Rabbit #A-21441) for 2 hours at room temperature. Nuclei were stained with DAPI (4′, 6-diamidino-2-phenylindole; Invitrogen, CA, USA). All samples were mounted with ProLong Diamond Antifade (Thermo. MA, USA) and applied for microscopy with Leica TCS SP8X confocal microscope. Images were analyzed with Leica Application Suite X.

### Antibodies

Sheep anti-Dig 488 (Roche, #1333089), mouse anti-Bio555 (Roche, #1297597) and rabbit anti-FITC (Molecular Probes, #A-889) were used in this study. All antibodies have been validated by the vendors and us for in situ FISH (https://www.science.org/doi/10.1126/science.1099247?url_ver=Z39.88-2003&rfr_id=ori:rid:crossref.org&rfr_dat=cr_pub%20%200pubmed).

### Statistics and reproducibility

Microsoft Excel 2019 (vl6.30) were used for data collection. The correlation analysis was performed with R using Pearson's correlation coefficient analysis. GraphPad Prism 8, R studio version 1.4.1717 were used for plotting and Illustrator (v24.0.1) was used for displaying. Images were analyzed with Leica Application Suite X. Fiji (OS version) and Photoshop (Photoshop CC v20.0.7) were used to adjust contrast and brightness of images, Helicon Focus (v7.6.1 Pro) was used to stack all images. GraphPad Prism 8 (v8.2.1) was used for data analysis and display. SnapGene (v5.0.7) was used for Sanger sequencing analysis. R studio (v4.1.0) was used for NGS data analysis. At least three single biological replicates were conducted for deep sequencing. All flies and mosquitoes were genotyped by scoring the fluorescence and then randomly selected for deep sequencing.

### Reporting summary

Further information on research design is available in the Nature Portfolio Reporting Summary linked to this article.

## Data availability

The sequences of *hth*CC plasmid used in this study have been deposited into GenBank Database under accession number OQ681082. All other plasmids refer to the publications Gerard et al., 2021 and Li et al., 2021. NGS raw sequencing data has been deposited at the NCBI Sequence Read Archive database under Bioproject PRJNA978340, PRJNA978619, PRJNA979933, PRJNA979941, PRJNA980914, PRJNA980915, PRJNA981558. Source data is provided in this paper as a Source Data File. Source data are provided with this paper.

## Code availability

R program code is available from the public Data Repository in GitHub [https://github.com/Zhiqian-Li/DSB-Classifier], https://doi.org/10.5281/zenodo.10655701[106].

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

## Acknowledgements

We are grateful to Annabel Guichard for assistance with fly genetics and offer particular thanks to David Kosman for conceiving the initial idea for developing the ICP analytic pipeline, building the R scripts and contributing to the design of the *hthCC* element and analysis of in situ activity in early embryos. Research was supported primarily by an award from the Bill & Melinda Gates Foundation to E.B and by R01 GM117321 (E.B.), R01 GM144608 (E.B.), R01 AI162911 (E.B.). All the NGS sequencing were conducted at the IGM Genomics Center, University of California, San Diego, La Jolla, CA. E.B. also receives support from the Tata Institute for Genetics and Society. This publication includes data generated at the UC San Diego IGM Genomics Center utilizing an Illumina NovaSeq 6000 that was purchased with funding from a National Institutes of Health SIG grant (#S10 OD026929).

## Author contributions

E.B. and Z.L. conceived the idea, Z.L. designed the construction, Z.L. analyzed the data and wrote the manuscript with contributions from the coauthors. L.Y. contributed to the mosquito experiments and library preparation. A.H. performed the in situ FISH. Z.L. designed all artwork and figures for the paper. All authors read and approved the final manuscript.

## Competing interests

E.B. has equity interests in Agragene Inc. and Synbal Inc., companies that may potentially benefit from the research results. E.B. also serves on the company's Board of Directors (Synbal) and Scientific Advisory Board (Synbal and Agragene). The terms of this arrangement have been reviewed and approved by the University of California, San Diego in accordance with its conflict-of-interest policies. All other authors declare no competing interests.
