## [Peer Review File · Nature Communications]

Developmental progression of DNA double-strand break repair
deciphered by a new single-allele resolution mutation
classifierReviewer #1 (Remarks to the Author):

“Developmental progression of DNA double-strand break repair deciphered by a new single-allele resolution mutation classifier”

Zhiqian Li, Lang You, Anita Hermann, Ethan Bier

In this manuscript, Li and colleagues utilize elegant genetics assays to induce DNA double-strand breaks (DSBs) followed by a novel robust bioinformatic approach, ICP (Integrated Classifier Pipeline), to analyze DSB repair outcomes (DSB fingerprint). They apply this new analysis at various loci and developmental times in *Drosophila* and *Anopheles*. In addition, they analyzed whether the parental source of Cas9 (maternal vs. paternal) impacted repair outcomes in the progeny.

Briefly, *Drosophila* containing Cas9 are crossed to flies containing guide RNAs (called “split” crosses), or Cas9 and gRNAs are derived from the same parent, (which they term “full”). Genomic DNA from F1 single progeny (or population of progeny in developmental experiments) is then amplified across the Cas9 break site. Deep sequencing followed by ICP analyses allow for characterization of non-HDR repair outcomes as Pam-End Proximal Protected Repair (PEPPR), MMEJ, deletion (DELET) or insertion (INSRT), resulting in a DSB “fingerprint” from each progeny sample. Analyses of repair events were highly reproducible between progeny, but varied considerably depending on the source of the Cas9 expression (i.e., maternally derived vs. paternally derived). Additionally, the source of Cas9 (i.e., *nanos*, *actin*, *vasa*), which impacts Cas9 expression, also resulted in different DSB fingerprints. These differences were consistent at different genomic loci within *Drosophila* as well as in *Anopheles*, suggesting a conservation of this phenomenon across insect species. The group analyzed DSB fingerprints of events occurring at different developmental time points in mosquitoes, by analyzing progeny from flies containing both Cas9 and guide RNAs (“full”) as embryos, larvae, pupae, and adults and found variation in DSB fingerprints depending on developmental time point. Similarly, when analyzing HDR repair events, they observed a difference in HDR frequency depending of the source of Cas9 (maternal vs. paternal) and developmental time point.

This is an exceptionally well-written paper with clear figures to follow the complex data sets that emerge from this novel analyses. Application of these elegant genetic tools allows for analysis of the impacts of the developmental timing of Cas9 expression, the level of Cas9 expression, and the source of Cas9 expression. This has significant implications in utilizing gene drives for insect control, other applications such as analyzing cancer cell progression in heterogenous cell populations, and more broadly to genome editing in general. This study underscores the variability of repair outcomes in complex multicellular organisms which has wide implications to the field of genome editing.

A few clarifying questions and/or minor suggestions to consider (particularly for those not well-versed in *Drosophila* genetics):

- 1) For general clarification for the maternal-F crosses (i.e., Figure 1), how is the experiment designed to prevent the gRNA expression to cross over and segregate away from the Cas9 source? Are the females balanced, the two loci tightly linked, homozygous, or do they rely on phenotypic markers in the F1 progeny to ensure both have segregated together? Similarly, for Fig. 6, maternal-F crosses followed by developmental analyses makes phenotypic scoring impossible in some stages (i.e. embryos and larvae). This point may be buried in the methods or previous published work, but it wasn't obvious and warrants clarification.
- 2) Figure 1 legend, that last sentence should be removed (“These rank-ordered fingerprints are highly reproducible between individuals”). This reproducibility is not shown until later figures (i.e., Fig 2) and this premature claim should be avoided at this point in the results.
- 3) Figure 3, it wasn't clear where the data in 3b were derived. Is this the same data from 2c? If not, what was the difference between these two data sets?

4) Figure 6, could the image-based clonal analysis also be a result of loss of heterozygosity, rather than HDR? Should this be included as a possible interpretation?

5) Throughout the manuscript, the authors refer to "gender" of the parental source of Cas9 as a factor in determining DSB fingerprints. As gender is a social construct, the term "sex" is more appropriate to use within a genetic context. I would recommend this change throughout.

6) Minor typo: Line 123, "Remaining alleles that including" should be "that include"

Reviewer #2 (Remarks to the Author):

This is a potentially interesting high throughput analysis pipeline to study DNA repair events that theoretically has wide application and could be a nice contribution to the DNA repair field. However, I have several major issues with the study that suggest a major rewrite is necessary to communicate their findings clearly and in the proper context of what's known about DNA repair (particularly in *Drosophila*).

1. The introduction overlooks critical information about double-strand break repair in metazoans. For example, they state on line 77 that "current mechanistic models of DSB repair have been informed to a large extent by studies performed in CRISPR/Cas9 treated mammalian cell lines using exogenously provided DNA repair templates...". The phrasing here suggests that most of what is known about DSB repair comes from Cas9 studies in cell culture, which overlooks the first 50 years of the entire field. Again, on line 92 the authors state "Therefore, analyzing DNA repair outcomes at fine scale with single-allele resolution within complex tissues composed of different cell types remains very challenging." There are decades of literature in metazoans, specifically in *Drosophila*, with these exact types of analyses. Because the introduction does not provide essential context, the authors unintentionally oversell the novelty of their findings.
2. As the authors note, a similar pipeline was recently published. While there is always room for more than one approach in science, the authors need to do a stronger job of communicating why their pipeline is novel and warrants publication in *Nature Communications*.
3. While the authors take their time to explain the analysis pipeline, they do not clearly explain the multiple different genetic systems they are using. The attempt in Figure 1 does not convey anything specific about the gene drive system. I would need to read previously published papers to understand the genetic system and therefore do not have enough information to assess the methods appropriately. That being said, at least one major concern about the methods is that the authors claim to be analyzing somatic DNA repair events, but from what I can tell, they are mostly using *vasa::Cas9* which is a germline specific driver. In *Drosophila*, *vasa::Cas9* has been shown to be leaky in somatic cells (<https://doi.org/10.1073/pnas.1405500111>), so at best, their results are from a combination of germline and somatic events. Additionally, *nanos::Cas9* is a very clean germline driver, so those results are purely germline. The data are interesting, but should be discussed in a framework of somatic vs. germline repair mechanisms.

Reviewer #3 (Remarks to the Author):

A major challenge with Cas9-mediated genome editing is predicting the most frequent types of repair events. Prior studies have demonstrated that cell cycle stage, genomic location of the break, and endonuclease source can all affect the spectrum of repair products, but how these factors impact repair in a multicellular organism is poorly understood. In this manuscript, Li et al. describe the development of a pipeline (ICP) to catalog and quantify repair events from NGS data. Using their classification scheme, they visualize rank-order classes of repair events in the form of color-coded DSB repair 'fingerprints.' Using the ICP, they show that the repair fingerprints are significantly different when the sgRNA/Cas9 are inherited from either male or female parents and that this difference is apparent in both *Drosophila* and *Anopheles*. To follow up on this, they conduct time course experiments with gene drive constructs in *Anopheles* and demonstrate that the repair fingerprints obtained from late, but not early, time points in development are very

different. Further experiments are performed to show the utility of the ICP in lineage tracing with multiple generations of large populations of insects. Finally, the authors show that they can follow homology-directed repair alongside end-joining solely by NGS analysis and that somatic HDR is very rare in *Drosophila* when the sgRNA/Cas9 is maternally inherited, but occurs at an appreciable frequency later in development with paternal sgRNA/Cas9 transmission.

Overall, there are several findings of interest in this paper. First, the ICP provides a highly visual way to compare the prevalence of different DNA repair events across different individuals. The use of allele fishing nicely allows for the inclusion of events that would normally be excluded based on commonly used alignment algorithms. Second, the ability to follow both HDR and end-joining repair using only an NGS platform is a useful advance for the field. Third, the observation of disparate repair fingerprints across different developmental stages in both flies and mosquitos is a novel finding. Finally, the developmental time courses strongly suggest that selection may cull detrimental repair events from multicellular organisms, a finding that is worthy of follow-up.

The methodology employed by the authors is appropriate, well-controlled, and fully described, although the descriptions are sometimes unclear for non-expert readers. I have moderate concerns about a lack of rationale for some repair event classification and potential misclassification using the ICP. I also think that some of the figures feature over-complicated data presentation. These concerns, described in detail below, should be addressed to maximize the impact of the study.

1. Inclusion of additional data:

- One of the most interesting results is the change in end-joining repair spectra in mosquitos throughout development (Fig 4). The authors investigate the balance of HR vs. indels in *Drosophila* during development in Fig. 6, but no analysis of how different classes of end joining repair events track during each stage is presented. Presumably the authors have these data from the NGS sequencing and including the analysis in a supplemental figure would nicely complement the Anopheles data.

2. ICP classification and representation of repair events:

- Some of the repair events appear to be misclassified. For example, in Fig 2a, repair events 2 and 3 from the Maternal-S cross could only be MMEJ if the Cas9 cut site is moved to the left 1 or 2 nt (unless I'm misunderstanding the way that MMEJ is determined).
- While I appreciate the color-coding approach for different classes of repair events, it wasn't clear to me why the authors created a class for deletions that only extended on the break side distal to the PAM (PEPPR). Presumably, the idea is that Cas9 that remains bound to the DNA following cleavage protects the PAM-proximal end from processing, but this is not stated, and it has been shown that Cas9 does interact with nucleotides on the PAM-distal side (<https://doi.org/10.1126/sciadv.aaw9807>). Is there a compelling reason not to code a class of deletions that occur only on the PAM proximal side? Were these recovered at a very low frequency?
- The authors choose to define MMEJ as ≥ 2 nt of microhomology, but pol-theta mediated end joining has been shown to operate with as little as 1 nt of microhomology in several organisms. They should justify their definition of MMEJ.
- In the third paragraph of the third results section, the authors mention that some of the alleles could be explained through a second round of repair using an end-joining repair event as a homologous template. This highlights the fact that repair classification using the ICP should not be interpreted as a proxy for repair mechanism. Other studies in *Drosophila* have also shown that presumed non-MMEJ deletion events can arise through an MMEJ pathway that involves multiple rounds of synthesis. This caveat should be emphasized in the discussion.
- In Fig 2a, there appears to be a discontinuity in Paternal-S repair event number 1. Is this a misprint?
- In Fig 3g, how are PEPPR mini, mini-I, midi-II and maxi defined?

3. Figure and data presentation: there are several instances where simplifying the figures could be helpful for the reader.

- In general, the use of colors is a nice feature but sometimes overdone. Various shades of blue are used for adenine, PEPPR repair, and maternal drive crosses. These can get confusing, especially when they appear next to each other.
- Fig 2C and 4G- The use of four different vertical levels for the class codes of the 50 most frequent repair events isn't necessary. Could these plots be flattened, as in the last panel in Fig. 1?
- Fig 2D,E - What is the advantage to showing plots for both allele class fraction and allele class rank index? Throughout the papers, these seem to show largely the same information. For me, the use of the non-intuitive class rank index added to the complexity and perhaps it could be omitted?
- Fig 4D - Is it necessary to show the percentage of both WT and indel repair events moving in opposition? Could showing the percentage of indel events for maternal-F vs. paternal-F be sufficient, if the text indicates that the other sequences were WT?

4 While I appreciate the authors' suggestion in the discussion that ICP could be useful in lineage analysis for gene drive experiments, it isn't clear to me how ICP will be applied to tumor metastasis or chromothripsis scenarios, when the precise location of double-strand breaks isn't known a priori.

Response to Reviewers' Comments

REVIEWER COMMENTS

Thank you again for submitting your manuscript "Deciphering developmentally regulated DNA double-strand break repair outcomes at single allele resolution in complex multicellular organisms" to Nature Communications. We have now received reports from 3 reviewers and, after careful consideration, we have decided to invite a major revision of the manuscript.

As you will see from the reports copied below, while the reviewers find your Integrated Classifier Pipeline (ICP) to be of potential interest, they also raise important concerns. In particular, the reviewers have requested clarification regarding the novelty of your method, and we suggest demonstrating its practical utility through additional examples. Please address all the reviewer's comments in your revision, while bearing in mind that we will be reluctant to approach the referees again in the absence of revisions.

If you feel that you are able to comprehensively address the reviewers' concerns, please provide a point-by-point response to these comments along with your revision. Please show all changes in the manuscript text file with track changes or colour highlighting. If you are unable to address specific reviewer requests or find any points invalid, please explain why in the point-by-point response.

We appreciate the constructive comments of the three reviewers, which we have addressed in the following point-by-point response. Accordingly, we have revised the corresponding manuscript text as indicated by changes highlighted in green.

Reviewer #1 (Remarks to the Author):

"Developmental progression of DNA double-strand break repair deciphered by a new single-allele resolution mutation classifier"

Zhiqian Li, Lang You, Anita Hermann, Ethan Bier

In this manuscript, Li and colleagues utilize elegant genetics assays to induce DNA double-strand breaks (DSBs) followed by a novel robust bioinformatic approach, ICP (Integrated Classifier Pipeline), to analyze DSB repair outcomes (DSB fingerprint). They apply this new analysis at various loci and developmental times in *Drosophila* and *Anopheles*. In addition, they analyzed whether the parental source of Cas9 (maternal vs. paternal) impacted repair outcomes in the progeny.

Briefly, *Drosophila* containing Cas9 are crossed to flies containing guide RNAs (called "split" crosses), or Cas9 and gRNAs are derived from the same parent, (which they term "full"). Genomic DNA from F1 single progeny (or population of progeny in developmental experiments) is then amplified across the Cas9 break site. Deep sequencing followed by ICP analyses allow for characterization of non-HDR repair outcomes as Pam-End Proximal Protected Repair (PEPPR), MMEJ, deletion (DELET) or insertion (INSRT), resulting in a DSB "fingerprint" from each progeny sample. Analyses of repair events were highly reproducible between progeny, but varied considerably depending on the source of the Cas9 expression (i.e., maternally derived vs. paternally derived). Additionally, the source of Cas9 (i.e., nanos, actin, vasa), which impacts Cas9 expression, also resulted in different DSB fingerprints. These differences were consistent at different genomic loci within *Drosophila* as well as in *Anopheles*, suggesting a conservation of this phenomenon across insect species. The group analyzed DSB fingerprints of events occurring at different developmental time points in mosquitos, by analyzing progeny from flies containing both Cas9 and guide RNAs ("full") as embryos, larvae, pupae, and adults and found variation in DSB fingerprints depending on developmental time point. Similarly, when analyzing HDR repair events, they observed a difference in HDR frequency depending of the source of Cas9 (maternal vs. paternal) and developmental time point.

This is an exceptionally well-written paper with clear figures to follow the complex data sets that emerge from this novel analysis. Application of these elegant genetic tools allows for analysis of the impacts of the developmental timing of Cas9 expression, the level of Cas9 expression, and the source of Cas9 expression. This has significant implications in utilizing gene drives for insect control, other applications such as analyzing cancer cell progression in heterogeneous cell populations, and more broadly to genome editing in general. This study underscores the variability of repair outcomes in complex multicellular organisms which has wide implications to the field of genome editing.

A few clarifying questions and/or minor suggestions to consider (particularly for those not well-versed in *Drosophila* genetics):

1) For general clarification for the maternal-F crosses (i.e., Figure 1), how is the experiment designed to prevent the gRNA expression to cross over and segregate away from the Cas9 source? Are the females balanced, the two loci tightly linked, homozygous, or do they rely on phenotypic markers in the F₁ progeny to ensure both have segregated together? Similarly, for Fig. 6, maternal-F crosses followed by developmental analyses makes phenotypic scoring impossible in some stages (i.e. embryos and larvae). This point may be buried in the methods or previous published work, but it wasn't obvious and warrants clarification.

Response: For all the split crosses in this study (Maternal-S or Paternal-S), we intercrossed homozygous strains of flies carrying gene cassettes encoding either Cas9 or the gRNA (except for the *pleCC* gRNA element which is inserted in *ple* gene on the third chromosome and is lethal when homozygous as a result of the insertion being a null-mutation of *ple* gene, as described in Li *et al.*, 2021). All F₁ progeny used for analyzing somatic DSB repair patterns carried both the Cas9 and gRNA transgenes, which could be genotyped by virtue of the different fluorescent markers they carried. This is also the case for Figure 6b using Maternal-S or Paternal-S crosses to detect the somatic HDR and INDELS in the same individual F₁ flies generated by the *pleCC* encoded gRNA. Although the same sample was used for quantification of both somatic HDR and INDEL events, the protocol we followed for genomic DNA analysis was modified for dual detection of HDR versus NHEJ events. Briefly, in the case of assessing somatic HDR events by NGS sequencing, we adapted our basic 2-step NGS library preparation protocol into a three-step polymorphism-based amplification scheme (see details in the Methods section). This protocol employs an insert-specific PCR primer and thus only amplifies genomic DNA carrying the inserted gene cassette located on either the donor chromosome or receiver chromosome (donor versus receiver alleles can be distinguished by closely linked alternative DNA polymorphisms). The result of this sequencing analysis is an estimate of the ratio of gene cassettes located on the receiver chromosomes versus total donor chromosomes (50% of all homologous chromosomes) which can be converted into the fraction of somatic HDR (fraction of HDR-converted receiver chromosome in total receiver chromosome) by the equation: (% Receiver reads/% Donor reads) * 100. For analysis of somatic INDELS in Figure 6b, the NGS libraries were prepared using the same basic 2-step scheme used in other experiments, in which case, all wild-type receiver chromosomes were amplified with short (200-400 bp) amplicons with the analysis providing an estimate for the ratio of WT to cut (and variously mutated) receiver chromosomes.

In the case of the Maternal-F and Paternal-F crosses presented in Figure 6c (and also Figure 6f), we performed a developmental analysis of somatic HDR by crossing a stock that was homozygous for *vasa*-Cas9 on the X chromosome and *pleCC* gRNA on the third chromosome over a dominantly marked balancer chromosomes (since homozygotes for this insertion are not viable), to wild type w¹¹¹⁸ flies. The donor chromosome segregated from the third chromosome balancer into the F₁ progeny. So, half of the F₁ progeny carry one allele of the balancer and one wild-type allele, these non-CRISPR progeny cannot contribute to analysis of somatic DSB repair frequencies, thus precluding this type of quantitative analysis in such a pooled experiment. Pooling is OK for analysis of somatic HDR since our protocol only amplifies alleles carrying the gene cassette. Thus, we pooled 20 F₁ progeny at different developmental stages for library preparation without a need for genotyping (i.e., the 50% of irrelevant non-CRISPR embryos are invisible to this protocol) to acquire data for averaged HDR events in the pool of 20 F₁ progeny embryos or larvae collected at different developmental stages as presented in Figs. 6c,f.

We were only able to perform a developmental analysis of the frequency and nature of NHEJ events in embryos carrying viable insertional alleles (e.g., Figs. 4d,e and newly added Supp. Fig. 9, for a gene cassette inserted into the *prosa2* locus that carried a recoded rescuing allele for the target gene). In these cases, we could only track indels events, but not HDR events, since we were unable to distinguish donor versus receiver polymorphisms, which is required for our NGS-based methods of HDR detection.

These various intricate points addressed above are now better clarified in the revised text and Methods sections.

2) Figure 1 legend, that last sentence should be removed ("These rank-ordered fingerprints are highly reproducible between individuals"). This reproducibility is not shown until later figures (i.e., Fig 2) and this premature claim should be avoided at this point in the results.

Response: We have removed this sentence accordingly.

3) Figure 3, it wasn't clear where the data in 3b were derived. Is this the same data from 2c? If not, what was the difference between these two data sets?

Response: The data presented in Figure 2c and 3b derived from different types of experiments (single flies versus pooled samples). Figure 2c is based on single fly sequencing data derived from Maternal-S versus Paternal-S crosses, while Figure 3b displayed averaged sequencing data generated from pooling 20 F₁ flies together for bulk sequencing. The fact that clearly distinct mutational fingerprints can be discerned from such bulk sequencing

comparisons in Fig. 3 a-d highlights how characteristic and reproducible mutational signatures are for a given cross.

4) Figure 6, could the image-based clonal analysis also be a result of loss of heterozygosity, rather than HDR? Should this be included as a possible interpretation?

Response: The image-based clonal analysis should not reflect loss of heterozygosity, since expression of DsRed fluorescence is only possible once the gene cassette has copied itself away from a closely linked ATG- mutation present on the donor chromosome onto the ATG+ receiver chromosome. Thus, expression of the fluorescent phenotypic marker indicates cells in which somatic HDR has taken place by copying the *p/eCC* construct to the receiver chromosome through HDR, as more fully analyzed in Li *et al.*, 2021.

5) Throughout the manuscript, the authors refer to “gender” of the parental source of Cas9 as a factor in determining DSB fingerprints. As gender is a social construct, the term “sex” is more appropriate to use within a genetic context. I would recommend this change throughout.

Response: We appreciate this point and have revised our use of this term accordingly throughout the manuscript.

6) Minor typo: Line 123, “Remaining alleles that including” should be “that include”

Response: We have corrected this sentence.

Reviewer #2 (Remarks to the Author):

This is a potentially interesting high throughput analysis pipeline to study DNA repair events that theoretically has wide application and could be a nice contribution to the DNA repair field. However, I have several major issues with the study that suggest a major rewrite is necessary to communicate their findings clearly and in the proper context of what’s known about DNA repair (particularly in *Drosophila*).

1. The introduction overlooks critical information about double-strand break repair in metazoans. For example, they state on line 77 that “current mechanistic models of DSB repair have been informed to a large extent by studies performed in CRISPR/Cas9 treated mammalian cell lines using exogenously provided DNA repair templates...”. The phrasing here suggests that most of what is known about DSB repair comes from Cas9 studies in cell culture, which overlooks the first 50 years of the entire field. Again, on line 92 the authors state “Therefore, analyzing DNA repair outcomes at fine scale with single-allele resolution within complex tissues composed of different cell types remains very challenging.” There are decades of literature in metazoans, specifically in *Drosophila*, with these exact types of analyses. Because the introduction does not provide essential context, the authors unintentionally oversell the novelty of their findings.

Response: We fully agree with the reviewer that it is important to summarize the major prior insights into the mechanisms of DSB repair that have been achieved using *Drosophila* as well as yeast, particularly as they pertain to elaboration of the components and roles of those repair pathways in meiosis. We have added a short section making that point in the introduction section and have added several key references representative of this large body of prior work:

1. Xue, C. and Greene, C.E. DNA repair pathway choices in CRISPR-Cas9-mediated genome editing. *Trends Genet.* 37, 639-656 (2021).
2. Haber, E.J. A life investigating pathways that repair broken chromosomes. *Annu. Rev. Genet.* 50, 1-28 (2016).
3. Jasin, M., Haber, E.J., The democratization of gene editing: insights from site-specific cleavage and double-strand break repair. *DNA Repair (Amst)*, 44, 6-16 (2016).
4. Sekelsky, J. DNA repair in *Drosophila*: mutagens, models, and missing genes. *Genetics*, 205, 471-490 (2017).
5. Gartner, A. and Engebrecht, J. DNA repair, recombination, and damage signaling. *Genetics*, 220, iyab178 (2022).
6. Davies, J.P., Evans, E.W. and Parry, M.J. Mitotic recombination induced by chemical and physical agents in the yeast *Saccharomyces cerevisiae*. *Mutat. Res.* 828, 111840 (1975).
7. Game, J.C. and Mortimer, R.K. A genetic study of x-ray sensitive mutants in yeast. *Mutat. Res.* 24, 281-292 (1974).
8. Liang, F. *et al.*, Chromosomal double-strand break repair in Ku80-deficient cells. *Proc. Natl. Acad. Sci. U S A.* 93, 8929-8933 (1996).

We have also revised the text in the introduction section to emphasize the novelty of our method which is to display categories of mutational events rather than raw sequence as data outputs thereby providing genetic fingerprints that reveal consistent and characteristic individual mutational patterns observed across several individual flies or mosquitoes that would be difficult, or impossible, to discern without this novel processing step. This "allelic category" versus "raw mutation" output is the key innovation and impactful application of our data analysis pipeline, which also includes integration with single cell resolution imaging techniques to detect and quantify HDR-mediated copying events across development of diverse somatic cell types. The ICP platform thus represents a substantial, not incremental, step forward in providing an unprecedented ability to distinguish even closely allied mutational fingerprints.

2. As the authors note, a similar pipeline was recently published. While there is always room for more than one approach in science, the authors need to do a stronger job of communicating why their pipeline is novel and warrants publication in Nature Communications.

Response: According to the reviewer's suggestion, we had added a more detailed comparison between ICP and Hussmann's (Hussmann *et al.*, 2021) pipeline in the first paragraph of the Discussion as well as Supplementary Discussion section. We also listed the major advantages of ICP when compared to their system.

1) In Hussmann's paper, the authors investigated the DSB repair pattern by transfecting mammalian cell lines with synthetic gRNA targets, rather than stable genomic DNA, which might not capture effects of genomic context and cell or tissue type on DSB repair outcomes. Our ICP platform can be flexibly adapted to distinguish distinct mutational signatures at several different endogenous genomic loci depending on various inheritance patterns of different Cas9 and gRNA expressing transgenes in the challenging context of complex multicellular organisms. 2) The Hussmann study used an iterative process to classify each UMI reads group and focused on the most common sequences in the group and discarded the reads with low consensus or low reads. However, with our system, virtually all reads could be analyzed as long as the dictionary includes all possible repair outcomes. 3) Hussmann *et al.* used a Knock-knock pipeline (Canaj *et al.*, 2019) to classify the DSB repair outcomes. Basically, it broke down the reads into few segments and generated a comprehensive set of local alignments between segments of the reads and all possible sequences present in the experiment. Their system then identified a subset of local alignments that covered the amplicon and parsed the arrangement of these alignments to classify the editing outcomes. Even if this method could be adapted to perform long-amplicon analysis, this segmented alignment complicates the reconstruction of whole reads, making it difficult to identify where read or sequencing errors may have occurred. In comparison, our ICP pipeline used a 24-nt seed region for classification, which is more straightforward and can more readily identify the vast majority of the technique-based (e.g. PCR or sequencing) errors at the same time. This is an important and innovative detail of our method. 4) Last, but most importantly, our ICP outputs an intuitively rank-ordered and color-coded fingerprints which is directly present the information about the most frequent alleles' frequency, category and even overall DSB repair patterns when compared with different experimental settings (e.g. diverse inheritance patterns of different CRISPR components). Such intuitive and informative mutational fingerprint outputs make it possible to track the origin of Cas9 or drive cassettes, and investigate dynamics of DSB repair pathways active during different stages of development.

Reference:

Canaj, G. *et al.*, Deep profiling reveals substantial heterogeneity of integration outcomes in CRISPR knock-in experiments. *bioRxiv*, 13 (2019).

We have added a short passage at the beginning of the Discussion section of the manuscript summarizing these key advantages of ICP platform.

3. While the authors take their time to explain the analysis pipeline, they do not clearly explain the multiple different genetic systems they are using. The attempt in Figure 1 does not convey anything specific about the gene drive system. I would need to read previously published papers to understand the genetic system and therefore do not have enough information to assess the methods appropriately. That being said, at least one major concern about the methods is that the authors claim to be analyzing somatic DNA repair events, but from what I can tell, they are mostly using *vasa::Cas9* which is a germline specific driver. In *Drosophila*, *vasa::Cas9* has been shown to be leaky in somatic cells (<https://doi.org/10.1073/pnas.1405500111>), so at best, their results are from a combination of germline and somatic events. Additionally, *nanos::Cas9* is a very clean germline driver, so those results are purely germline. The data are interesting, but should be discussed in a framework of somatic vs. germline repair mechanisms.

Response: We appreciate this point since it is important to clarify the distinction between our prior studies where we concentrated on germline transmission of gene-drive elements versus the focus in this current study on somatic editing events (except for Figure 6 in which we do use NGS sequencing together with our prior described imaged based CopyCatcher detection system to quantify somatic HDR events). We revised the manuscript in the Results

section to clarify this point and have elaborated on details of the various genetic arrangements more fully in the Methods section. Briefly, we conducted two split-drive crosses (Maternal-S and Paternal-S) and two full-drive crosses (Maternal-F and Paternal-F). For the split crosses the two components of Cas9 system (Cas9 protein and gRNA constructs) were inherited separately from their parents, while for the full-drive crosses a single gene cassette carrying the linked Cas9 and gRNA elements was inherited as a single unit.

With regard to the germline versus somatic activities of the *vasa::Cas9* and *nanos::Cas9*, we and other groups have observed significant levels of leaky expression of these transgenes in somatic cells especially for *vasa::Cas9* (Gantz and Bier, 2015, Hammond *et al.*, 2016, Champer *et al.*, 2018). Even the *nanos* promoter generates a limited degree of somatic Cas9 activity such as that analyzed in Figure 3d. Since we focused our analysis only on the events scored in the F₁ generation (except for the data presented in Figure 5 that includes multi-generational cage experiments), and because meiotic cells make up only a very small proportion of the total cells in an adult fly or mosquito, our data are dominated by the somatic activities of the gene cassettes.

We have clarified these important points in the revised text.

Reviewer #3 (Remarks to the Author):

A major challenge with Cas9-mediated genome editing is predicting the most frequent types of repair events. Prior studies have demonstrated that cell cycle stage, genomic location of the break, and endonuclease source can all affect the spectrum of repair products, but how these factors impact repair in a multicellular organism is poorly understood. In this manuscript, Li *et al.* describe the development of a pipeline (ICP) to catalog and quantify repair events from NGS data. Using their classification scheme, they visualize rank-order classes of repair events in the form of color-coded DSB repair 'fingerprints.' Using the ICP, they show that the repair fingerprints are significantly different when the sgRNA/Cas9 are inherited from either male or female parents and that this difference is apparent in both *Drosophila* and *Anopheles*. To follow up on this, they conduct time course experiments with gene drive constructs in *Anopheles* and demonstrate that the repair fingerprints obtained from late, but not early, time points in development are very different. Further experiments are performed to show the utility of the ICP in lineage tracing with multiple generations of large populations of insects. Finally, the authors show that they can follow homology-directed repair alongside end-joining solely by NGS analysis and that somatic HDR is very rare in *Drosophila* when the sgRNA/Cas9 is maternally inherited, but occurs at an appreciable frequency later in development with paternal sgRNA/Cas9 transmission.

Overall, there are several findings of interest in this paper. First, the ICP provides a highly visual way to compare the prevalence of different DNA repair events across different individuals. The use of allele fishing nicely allows for the inclusion of events that would normally be excluded based on commonly used alignment algorithms. Second, the ability to follow both HDR and end-joining repair using only an NGS platform is a useful advance for the field. Third, the observation of disparate repair fingerprints across different developmental stages in both flies and mosquitos is a novel finding. Finally, the developmental time courses strongly suggest that selection may cull detrimental repair events from multicellular organisms, a finding that is worthy of follow-up.

The methodology employed by the authors is appropriate, well-controlled, and fully described, although the descriptions are sometimes unclear for non-expert readers. I have moderate concerns about a lack of rationale for some repair event classification and potential misclassification using the ICP. I also think that some of the figures feature over-complicated data presentation. These concerns, described in detail below, should be addressed to maximize the impact of the study.

1. Inclusion of additional data:

- One of the most interesting results is the change in end-joining repair spectra in mosquitos throughout development (Fig 4). The authors investigate the balance of HR vs. indels in *Drosophila* during development in Fig. 6, but no analysis of how different classes of end joining repair events track during each stage is presented. Presumably the authors have these data from the NGS sequencing and including the analysis in a supplemental figure would nicely complement the *Anopheles* data.

Response: In Figure 6b, we used the *pleCC* gRNA bearing element for analysis of both somatic HDR and NHEJ, which is a follow-up experiment to our previous studies using *pleCC* phenotypically quantifying somatic HDR (Li *et al.*, 2021). The *pleCC* gRNA cassette was inserted into the first intron of *ple* gene and generates a lethal null mutation. Because homozygous *ple* null mutation are inviable, we kept this stock with a balancer on the third chromosome. Thus, crosses with this line will produce F₁ progeny half of which are not suitable for somatic INDEL analysis (i.e., those carrying the balancer). When analyzing data for individual flies this is possible since we can choose which flies

to sequence based on the presence or absence of fluorescence markers. In contrast, for our developmental analysis where we needed to pool ~20 embryos, it is not possible to genotype individual progeny particularly during embryonic and young larval stages. Thus, to explore the developmental profile of somatic INDELS we chose another gene target (*prosa2*) for which the gRNA is carried on a gene cassette with a recoded target locus permitting this element to survive in a homozygous condition and thus permitting all embryos analyzed to carry the relevant gene cassette. This new piece of data has been added as supplementary Figure 9. Notably, data obtained from the *prosa2* gRNA bearing cassette only permitted developmental analysis of somatic INDELS, and not HDR analysis due to the absence of required sequence polymorphisms tightly linked to the gRNA cut site that could be used for distinguishing donor versus copied receiver chromosomes (see Methods section for detailed information). Thus, although we did not have a single perfect system for detecting both INDELS and HDR in somatic cells of embryos and larvae, we were able to obtain these types of data separately and in the case of adult flies could capture both types of events in the same individuals.

2. ICP classification and representation of repair events:

- Some of the repair events appear to be misclassified. For example, in Fig 2a, repair events 2 and 3 from the Maternal-S cross could only be MMEJ if the Cas9 cut site is moved to the left 1 or 2 nt (unless I'm misunderstanding the way that MMEJ is determined).

Response: The reviewer is correct in bringing up this point. Regarding alleles 2 and 3 from the Maternal-S cross, they might have resulted from MMEJ repair if Cas9 created a staggered cut, rather than the more common blunt cleavage (thereby moving the cut site to the left 1 or 2 nt). Based on previous publications (e.g., Shi *et al.*, 2019 in HEK293T cells, Gisler *et al.*, 2019 in mouse pre-B cells, Shou *et al.*, 2018 in HEK293T cell) which identified the large majority of primary Cas9 cut sites between the 6th and 7th nucleotide position (counted from the last nucleotide of PAM), we only considered this simplifying situation in our analysis.

We now explicitly mention this assumption and approximation in the revised text.

Relevant added References :

- 1) Shi, X. *et al.*, Cas9 has no exonuclease activity resulting in staggered cleavage with overhangs and predictable di- and tri-nucleotide CRISPR insertions without template donor. *Cell Discov*, 5, 53 (2019).
- 2) Gisler, S. *et al.* Multiplexed Cas9 targeting reveals genomic location effects and gRNA-based staggered breaks influencing mutation efficiency. *Nat. Commun.* 10, 1598 (2019).
- 3) Shou, J., Li, J, Liu, Y and Wu, Q. Precise and predictable CRISPR chromosomal rearrangements reveal principles of Cas9-mediated nucleotide insertion. *Mol. Cell.* 71, 498-509 (2018).

- While I appreciate the color-coding approach for different classes of repair events, it wasn't clear to me why the authors created a class for deletions that only extended on the break side distal to the PAM (PEPPR). Presumably, the idea is that Cas9 that remains bound to the DNA following cleavage protects the PAM-proximal end from processing, but this is not stated, and it has been shown that Cas9 does interact with nucleotides on the PAM-distal side (<https://doi.org/10.1126/sciadv.aaw9807>). Is there a compelling reason not to code a class of deletions that occur only on the PAM proximal side? Were these recovered at a very low frequency?

Response: The PAM-proximal end protected category of DSB repair or PEPPR as we termed in this study has been well characterized in prior studies including those of Wienert *et al.*, 2019 and Richardson *et al.*, 2016. Since we similarly observed PEPPR alleles as a dominant class in all the tested targets, we created a specific class for this frequent sub-category of events.

References:

- 1) Wienert, B. *et al.*, Unbiased detection of CRISPR off targets *in vivo* using DISCOVER-seq. *Science*, 364, 286-289 (2019).
- 2) Richardson, D.C, Ray, J.G., DeWitt, A.M., Curie, L.G. and Corn, E.J. Enhancing homology-directed genome editing by catalytically active and inactive CRISPR-Cas9 using asymmetric donor DNA. *Nat. Biotechnol.* 34, 339-344 (2016).

- The authors choose to define MMEJ as ≥ 2 nt of microhomology, but pol-theta mediated end joining has been shown to operate with as little as 1 nt of microhomology in several organisms. They should justify their definition of MMEJ.

Response: It is true that the MMEJ DSB repair pathway can employ only 1 nt of microhomology for DSB repair in mammalian cells (Simsek and Jasin. 2010) and *Caenorhabditis elegans* (Koole et al., 2014). Although ignoring this minority class of events does lead to an underestimate of MMEJ alleles and the failure to distinguish this subset of MMEJ events from other categories, we found it to be too demanding computationally to define these events in a robust fashion and thus decided to place the threshold at ≥ 2 nt. This was also the case for Hussmann's study where they too used a ≥ 2 nt match for the criteria to define the MMEJ category.

References:

- 1) Simsek, D. and Jasin, M. Alternative end-joining is suppressed by the canonical NHEJ component Xrcc4-ligase during chromosomal translocation formation. *Nat. Struct. Mol. Biol.* 17, 410-416 (2010).
- 2) Koole, W. *et al.*, A polymerase Theta-dependent repair pathway suppresses extensive genomic instability at endogenous G4 DNA sites. *Nat. Commun.* 5, 3216 (2014).
- 3) Chan, H.S., Yu, M.A. and McVey, M. Dual roles for DNA polymerase theta in alternative end-joining repair of double-strand breaks in *Drosophila*. *PLoS Genet.* 6, e1001005 (2010).

- In the third paragraph of the third results section, the authors mention that some of the alleles could be explained through a second round of repair using an end-joining repair event as a homologous template. This highlights the fact that repair classification using the ICP should not be interpreted as a proxy for repair mechanism. Other studies in *Drosophila* have also shown that presumed non-MMEJ deletion events can arise through an MMEJ pathway that involves multiple rounds of synthesis. This caveat should be emphasized in the discussion.

Response: This is a good point. We have added discussion of this alternative potential generation of non-MMEJ deletion events and cite corresponding references in this regard.

References:

- 1) Yu, M.A. and McVey, M. Synthesis-dependent microhomology-mediated end joining accounts for multiple types of repair junctions. *Nucleic Acids Res.* 38, 5706-5717 (2010).
- 2) Ramsden A.D., Carvajal-Garcia, J. and Gupta, P.G. Mechanism, cellular functions and cancer roles of polymerase-theta-mediated DNA end joining. *Nat. Rev. Mol. Cell Biol.* 23, 125-140 (2022).

- In Fig 2a, there appears to be a discontinuity in Paternal-S repair event number 1. Is this a misprint?

Response: Yes, there is a misprint in allele 1 of Paternal-S repair in Fig. 2a and we have corrected it.

- In Fig 3g, how are PEPPR mini, mini-I, midi-II and maxi defined?

Response: The definition of PEPPR mini, mini-I, midi-II and maxi was based on the length of deletion. The averaged deletion size were 24 bp, 40 bp, 31 bp for PEPPR-Mini, Mini-I and Midi-II cluster, and for PEPPR-Maxi cluster, it was longer than 55 bp. We now explicitly specify these threshold ranges in the text.

3. Figure and data presentation: there are several instances where simplifying the figures could be helpful for the reader.

- In general, the use of colors is a nice feature but sometimes overdone. Various shades of blue are used for adenine, PEPPR repair, and maternal drive crosses. These can get confusing, especially when they appear next to each other.

Response: We have changed the color of axis labels in figure 2c-f, figure 3i-l, and figure 4b to black to simplify these schemes and to avoid such confusion. Also, we changed the color of lines showing the indels profile with Maternal-S crosses from sky-blue into dark-blue, as well as the side bars in the fingerprints into the corresponding color, to distinguish the color codes for crosses from that showed the nucleotides.

- Fig 2C and 4G— The use of four different vertical levels for the class codes of the 50 most frequent repair events isn't necessary. Could these plots be flattened, as in the last panel in Fig. 1?

Response: We have followed the reviewer's suggestion and have compressed the fingerprint plots into single rows in Figure 2C and Figure 4G.

- Fig 2D,E - What is the advantage to showing plots for both allele class fraction and allele class rank index? Throughout the papers, these seem to show largely the same information. For me, the use of the non-intuitive class rank index added to the complexity and perhaps it could be omitted?

Response: We agree with the reviewer and have removed the class rank in all figures.

- Fig 4D – Is it necessary to show the percentage of both WT and indel repair events moving in opposition? Could showing the percentage of indel events for maternal-F vs. paternal-F be sufficient, if the text indicates that the other sequences were WT?

Response: We have revised the figure 4D accordingly, into showing indels' frequency only.

4 While I appreciate the authors' suggestion in the discussion that ICP could be useful in lineage analysis for gene drive experiments, it isn't clear to me how ICP will be applied to tumor metastasis or chromothripsis scenarios, when the precise location of double-strand breaks isn't known a priori.

Response: In this study, we employed our ICP platform to parse DSB repair outcomes on targeted cleavage sites. For less well-defined targets such those that could be used for tracking tumor metastasis, chromothripsis or off-targets, we now elaborate in the discussion section that we would define candidate mutational hotspots or off-target sites by using a variety of methods such as Circle-seq or DISCOVER-seq. Specifically, DISCOVER-seq performed by CHIP-seq with antibody against the core DNA repair factors (MRE11) that being recruited to the double-strand breaks after the CRISPR/Cas9 cleavage. When coupled with ICP analysis, DISCOVER-seq provides information regarding the location of Cas9 bound off-target sequences including those that could be used as the reference sequences for ICP analysis (Kim et al., 2015, Wienert et al., 2019, Tsai et al., 2017). Similarly, ICP analysis could potentially contribute to defining and assessing categories of events occurring at candidate mutational hotspots in certain genetic conditions (e.g., fragile chromosome syndromes) or primary versus developing tumors by performing Circle-seq to identify signature recurrent mutations such as those bordering genome rearrangements. Allelic dictionaries could thus be constructed to follow the occurrence and nature of such relevant recurrent alleles generated during dynamic cancer progression at single-allele resolution by taking tissue biopsies at different stages of tumor progression.

We elaborate on these points in the revised Discussion section.

References:

- 1) Kim, D. *et al.*, Digenome-seq: genome-wide profiling of CRISPR-Cas9 off-target effects in human cells. *Nat. Methods*, 12, 237-243 (2015).
- 2) Wienert, B. *et al.*, Unbiased detection of CRISPR off-targets in vivo using DISCOVER-Seq. *Science*, 364, 286-289 (2019).
- 3) Tsai, Q.S. *et al.*, CIRCLE-seq: a highly sensitive in vitro screen for genome-wide CRISPR-Cas9 nuclease off-targets. *Nat. Methods*, 14, 607-614 (2017).

Reviewer #1 (Remarks to the Author):

The revised manuscript has addressed my concerns. I appreciate the robust response to other reviewer's comments as well. This revision has improved the manuscript and I recommend accepting for publication.

Reviewer #2 (Remarks to the Author):

The authors have addressed the issues well and I have no other remaining problems with the manuscript.

Reviewer #2 (Remarks on code availability):

Code is present at the url.

Reviewer #3 (Remarks to the Author):

The authors have sufficiently addressed all my concerns in their revised manuscript. Specifically, they have revised the figures for clarity and have included methodological statements in the main text that clarify their experimental design. I appreciate their inclusion of another locus for the *Drosophila* time course experiment (Sup. Fig. 9). It would be useful to compare these time course results more directly for mosquito and fly, beyond the somewhat cryptic "...A consistent result was also observed with fly *prosa2* gRNA, even the DSB repair patterns are distinct" that is currently in the results section.

Two final minor points:

1. Line 569: "...by using antibody against to DSB repair core factors (MRE11)" should be rewritten.
2. The final paragraph in the "Limitations of the study" section is not about limitations of this study, but rather extensions of the ICP that can address limitations of other studies. It should be moved to the earlier discussion section.

Reviewer #3 (Remarks on code availability):

I did not install and run the code provided in the github directory, although I did look through the code. There does not appear to be a README file for novice users. I do think that the code could be executed by users that are experienced with sequence analysis.

Response to reviewer comments:

Both Reviewers 1 and 2 were satisfied with the manuscript revisions. Reviewer 3 had several additional comments which we address here:

The authors have sufficiently addressed all my concerns in their revised manuscript. Specifically, they have revised the figures for clarity and have included methodological statements in the main text that clarify their experimental design. I appreciate their inclusion of another locus for the *Drosophila* time course experiment (Sup. Fig. 9). It would be useful to compare these time course results more directly for mosquito and fly, beyond the somewhat cryptic "...A consistent result was also observed with fly *prosa2* gRNA, even the DSB repair patterns are distinct" that is currently in the results section.

We have clarified this interesting point as suggested in the final revised manuscript:

"Similarly, a split-drive expressing a gRNA targeting the *Drosophila prosaA* locus generated distinct temporal profiles of cleavage patterns in crosses from female versus male parents carrying the drive element (Fig. S9)."

Two final minor points:

1. Line 569: "...by using antibody against to DSB repair core factors (MRE11)" should be rewritten.

We have rephrased this passage as follows:

"Similarly, ICP analysis could potentially contribute to defining and assessing categories of events occurring at candidate mutational hotspots in certain genetic conditions (e.g., fragile chromosome syndromes) or primary versus developing tumors by performing CHIP-seq by using antibodies against to DSB repair core factors (e.g., MRE11). Such an analysis might identify signature recurrent mutations such as NHEJs bordering genome rearrangements due to cleavage and inaccurate rejoining of broken ends from two different chromosomes (or inversions within the same chromosome)."

2. The final paragraph in the "Limitations of the study" section is not about limitations of this study, but rather extensions of the ICP that can address limitations of other studies. It should be moved to the earlier discussion section.

We have moved this final paragraph to the first section of the Discussion as suggested.